# Partially Observed Trajectory Inference using Optimal Transport and a Dynamics Prior

**Anming Gu & Edward Chien**
Boston University
Boston, MA
{agu2002, edchien}@bu.edu

**Kristjan Greenewald**
MIT-IBM Watson AI Lab; IBM Research
Cambridge, MA
kristjan.h.greenewald@ibm.com

## Abstract

Trajectory inference seeks to recover the temporal dynamics of a population from snapshots of its (uncoupled) temporal marginals, i.e. where observed particles are *not* tracked over time. Prior works addressed this challenging problem under a stochastic differential equation (SDE) model with a gradient-driven drift in the observed space, introducing a minimum entropy estimator relative to the Wiener measure and a practical grid-free mean-field Langevin (MFL) algorithm using Schrödinger bridges. Motivated by the success of observable state space models in the traditional paired trajectory inference problem (e.g. target tracking), we extend the above framework to a class of latent SDEs in the form of *observable state space models*. In this setting, we use partial observations to infer trajectories in the latent space under a specified dynamics model (e.g. the constant velocity/acceleration models from target tracking). We introduce the PO-MFL algorithm to solve this latent trajectory inference problem and provide theoretical guarantees to the partially observed setting. Experiments validate the robustness of our method and the exponential convergence of the MFL dynamics, and demonstrate significant outperformance over the latent-free baseline in key scenarios.

## 1 Introduction

Estimating the temporal dynamics and trajectories of a population from collections of unpaired observations[1] at specific time points is a challenging fundamental problem with many potential applications such as single-cell genomic data analysis (Lavenant et al., 2024; Chizat et al., 2022). Existing work has focused on trajectory inference in the *fully observed setting*, where all variables that are important to the underlying dynamics are directly observed with no hidden states. We note that research in signal processing and control theory has overwhelmingly shown the importance of being able to handle latent states in dynamics modeling (Hespanha, 2018; Davis, 2013), since the best physical predictive quantities are often unobserved (for instance velocity and acceleration of an object whose positions are observed). Even linear state space models have enjoyed a recent resurgence for modeling text sequence data with large language models (Gu & Dao, 2024).

While in general the problem of recovering a hidden state is not identifiable, (Kalman, 1960) etc. introduced systems theory and developed observability conditions on the underlying dynamics model that does allow for such recovery. For instance, in target tracking (Doucet et al., 2001; Bar-Shalom & Li, 1995), oftentimes only a position variable is observed, yet the tracking algorithm uses a state space model that includes a hidden velocity state. This hidden state allows for better predicting the future position of the target, improving the trajectory inference by not only better modeling the dynamics, but making it easier to identify which of several targets observed at any given time correspond to the current track. The hidden states themselves are also often of direct interest.

The trajectory inference problem has many similarities to the tracking problem. In particular, at any given time, a cloud of points is observed, but these points are not labeled, i.e. there is no indication of which points at time $t_1$ match to the points at time $t_2$. Inferring these "matches" is the task of trajectory inference, and closely parallels the data association problem described by (Kirubarajan

---

[1]This corresponds to distributions being independent, e.g. see (iv) of Theorem C.1.

& Bar-Shalom, 2004; Rezatofighi et al., 2015) in target tracking. As a result, we are motivated to introduce latent state space modeling to the trajectory inference problem.

While itself being a fully observed framework without incorporating a known dynamics, the optimal transport (OT)-based method of (Chizat et al., 2022) is particularly amenable to our aims. It proposes to optimize collections of particles at each time step to minimize a data fit term (a cost between the particle cloud and the observed data points) and a trajectory fit term consisting of the entropic Wasserstein distance between sets of particles at adjacent time points. The entropic OT framework arises naturally from the SDE model (as we will see later), and provides an explicit and robust procedure for obtaining inferred trajectories from unpaired time series data by following the OT plan between time points. Representing the inferred time marginal densities as particles is also particularly amenable to our partially observed framework, as we can have the particles be in the hidden state space and form a data fit term to the observations using a specified (stochastic) observation model. In many ways this parallels the observation model/hidden state particle setup used by the particle filter (Arulampalam et al., 2002) and other sequential Monte Carlo methods (Doucet et al., 2001) for the paired-observation trajectory inference setting.

While our focus is theoretical, we envision the latent trajectory inference problem being broadly applicable to a variety of real-world tasks. For instance, the extra smoothing induced by introducing a hidden velocity state could improve trajectory inference directly in many settings, including possibly the genomic data analysis setting mentioned by (Chizat et al., 2022). Another application domain could be survey or medical data. Specifically, when trajectory data is needed, *longitudinal studies* are often designed where individuals are followed over time and continue to be re-interviewed, but significant logistical challenges are involved in such a procedure (Thomson & Holland, 2003). Trajectory inference could allow for different individuals to be sampled at each time point, significantly easing the burden on researchers. Our approach for introducing hidden states (e.g. velocity) would be particularly impactful, as in both social science and medical data, often individual's trajectories (e.g. preferences or health) do carry significant momentum. Finally, our approach could have advantages in private learning of time series models or private synthetic data generation for time series. This is because in (differential) privacy (Dwork et al., 2006; Dwork & Roth, 2014), the goal is to preserve the statistics of an individual, and trajectory inference allows for each individual's data to be limited to a single point in time, not requiring a full trajectory record that may be difficult to privatize. Our partial observation framework would allow for even more privacy to be maintained as some variables could remain hidden if an appropriate dynamics model was available.

Note the trajectory inference problem is much more difficult than simply tracking/estimating the sequence of state distributions at each time point (e.g. (Kim et al., 2021)). While these time marginals can be extracted from the inferred trajectory distribution, the inferred trajectory distribution is a *joint* distribution over time, i.e. providing *couplings* between the distributions at each time point. Crucially, this coupling allows for the *sampling* of trajectories from the learned distribution and implies knowledge of individual dynamics, rather than simply uncoupled group dynamics.

**Related works** (Saelens et al., 2019) surveys and compares of single-cell trajectory inference methods. (Schiebinger et al., 2019) introduces the use of OT for trajectory inference; however, the method generates paths that are generally not smooth. (Chewi et al., 2021) uses OT to construct measure-valued splines, which yields smooth paths, (Bunne et al., 2022) models population dynamics as a Jordan-Kinderlehrer-Otto (JKO) flow, and (Qu et al., 2024) uses OT to analyze gene trajectories.

(Weinreb et al., 2018) consider the limits of trajectory inference from single-cell snapshots in the equilibrium setting. However, as far as we are aware, (Lavenant et al., 2024) was the first work to provide theoretical guarantees of any estimator for trajectory inference. They introduce a min-entropy estimator for unknown gradient-driven drift models and prove convergence to the ground truth in the limit as the number of observations become dense in the observation period. (Chizat et al., 2022) extends the entropy minimization formulation of (Lavenant et al., 2024) by considering a different fitting functional, reducing optimization space, and using MFL dynamics. (Zhang et al., 2021) considers the application of these OT frameworks to the steady-state setting with known cell birth and death rates. A recent work (Ventre et al., 2024) builds on (Lavenant et al., 2024) and provides theoretical guarantees for trajectory inference in the branching case. Similar in spirit to our work, Shen et al. (2024) consider trajectory inference in the setting where a family of reference measures is specified, and Zhang et al. (2025) considers learning the unbalanced setting via neural networks. (Kim et al., 2021) considers a similar latent space dynamics model as us and aims to

recover the time series of a latent population distribution, but as noted above their work does not infer a trajectory distribution (specifically, the inferred time marginals are not coupled), and therefore is not designed to generate trajectory samples or characterize the individual dynamics (as opposed to population dynamics). These works, along with ours, do not aim to recover the drift dynamics; however, the works (Bunne et al., 2022; Tong et al., 2020; Hashimoto et al., 2016) do study this problem.

(Vahdat et al., 2021) uses score-based generative modeling in latent space. (Jiao et al., 2024) uses a pre-trained encoder and decoder, consider diffusion in latent space, and prove theoretical guarantees that the output distribution is close to the ground truth. (Hamm et al., 2024; Zhang et al., 2023) consider learning latent manifold structures using OT, Song et al. (2023b) consider gradient flow in latent space to study equivariant networks, (Song et al., 2023a) study the latent space of generative models using OT, and (Al-Jarrah et al., 2024) consider the nonlinear filtering problem using partial observations using OT. Others consider applications of Schrödinger bridges with non-Wiener reference measures. For example, (Chen et al., 2016) consider Schrödinger bridges where the prior is any Markov evolution for control theory and (Bunne et al., 2023) show that Schrödinger bridges between Gaussians against reference measures induced by linear SDEs have a closed forms.

**Notation** For probability measures $\mu$, $\nu$, the relative entropy (e.g. the KL divergence) is $H(\mu|\nu) = \int \log(d\mu/d\nu)d\mu$ if $\mu \ll \nu$ and $+\infty$ otherwise. For $n \in \mathbb{N}$, let $[n] := \{1, \ldots, n\}$. We use $\mathcal{X}$ to denote the *latent* space and $\mathcal{Y}$ to denote the *observation* space. We use the notation $\mathcal{P}(\cdot)$ to denote the probability distributions over a space. The path space is $\Omega = C([0, 1] : \mathcal{X})$, the set of continuous $\mathcal{X}$-valued paths. In our theoretical discussion, as in (Lavenant et al., 2024), we assume without loss of generality that the end time interval is $t = 1$. If $\mathbf{R} \in \mathcal{P}(\Omega)$ is a probability measure on the space of paths, its marginal at time $t$ is denoted as $\mathbf{R}_t \in \mathcal{P}(\mathcal{X})$. We generally use the Greek letters $\mu, \rho$ to denote probability distributions on $\mathcal{X}, \mathcal{Y}$, respectively. We use $\delta_x$ to denote a Dirac delta at $x$. By an abuse of notation, we use $|\cdot|^2 = \langle \cdot, \cdot \rangle \in \mathbb{R}$ for both the squared norm of a vector and the quadratic variation of a stochastic process. For a function $g : \mathcal{X} \to \mathcal{Y}$ and a measure $\mu$, we use $g_\sharp \mu$ to denote the pushforward measure, e.g. $g_\sharp \mu(B) = \mu(g^{-1}(B))$ for a measurable set $B \subseteq \mathcal{Y}$.

## 2 LATENT TRAJECTORY INFERENCE

We now state the problem of latent trajectory inference, which informally is depicted in Figure 1. At a high level, trajectories are modeled as being generated from a stochastic differential equation model (SDE) which incorporates both known terms (e.g. the latent dynamics model describing known relationships between states[2]) and unknown terms that allow for unknown forces or model misspecification. Note that specifying the dynamics between the latent states is crucial to make unobserved states *identifiable* as discussed later. Specifically, let $X_t \in \mathcal{X}$ be an unobserved state vector evolving according to the following SDE for $t \in [0, 1]$:

$$dX_t = -\Xi(t, X_t)dt - \nabla\Psi(t, X_t)dt + \sqrt{\tau}dB_t, \tag{1}$$

where $\{B_t\}$ is a Brownian motion, $\tau$ is the *known* diffusivity parameter, $\Xi \in C([0, 1] \times \mathcal{X} : \mathcal{X})$ is a *known* driving vector field (i.e. the dynamics model), and $\Psi \in C^2([0, 1] \times \mathcal{X})$ is an *unknown* potential function. Let $\mathbf{P}$ be the law of the SDE with initial condition $\mathbf{P}_0$ where $\mathbf{P}_t \in \mathcal{P}(\mathcal{X})$ are the marginals of $\mathbf{P}$ at time $t \in [0, 1]$. Our SDE differs from that of (Lavenant et al., 2024; Chizat et al., 2022) by the addition of non-zero known $\Xi$ which allows for idenfiable non-observed latent states.

As this latent space is not observed, we require a model mapping the latent space to the observation space in which our data samples live. Consider a smooth function $g : \mathcal{X} \to \mathcal{Y}$ transforming $X_t$ into the observation space $\mathcal{Y}$: $Y_t = g(X_t)$. Suppose we have $T$ observation times with $0 \leq t_1^T < \cdots < t_T^T \leq 1$, and we observe $N_i^T$ i.i.d. samples from the marginal distribution of $Y_{t_i^T}$:

$$\{Y_{i,j}^T\}_{j=1}^{N_i^T} \overset{\text{i.i.d.}}{\sim} g_\sharp \mathbf{P}_{t_i^T} := \mathbf{Q}_{t_i^T}, \tag{2}$$

forming empirical distributions $\hat{\rho}_i^T = \frac{1}{N_i^T} \sum_{j=1}^{N_i^T} \delta_{Y_{i,j}^T}$. Here, note that our framework allows for varying number of samples at each time point: $i$ indexes the time, $j$ indexes the data points, and the $T$ superscript is used to highlight the dependence on the number of time points.

---

[2]For instance, the fact that a velocity state additively impacts future positions states.

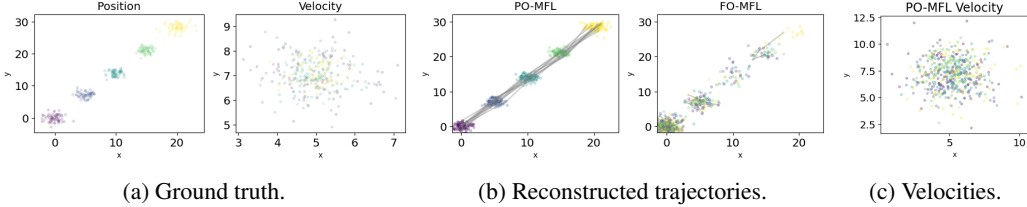

Figure 1: Constant velocity model, where the variance of the ground truth has been rescaled. We see that our method, PO-MFL, is more robust as the baseline method, FO-MFL, fails to converge, and provides per-particle velocity in contrast to FO-MFL. See Section 4 for the experiment setting.

Our goal is to recover the trajectory distribution $\mathbf{P}$ given the marginal snapshots in *observation space* $(\hat{\rho}_1^T, \ldots, \hat{\rho}_T^T)$. In particular, we provide an algorithm to sample trajectories in the *latent space* (which can then be mapped to trajectories on the observation space). In general, to make this problem well-posed and tractable, we make several assumptions on this very general setup. The first of these is a notion of observability that generalizes the ensemble observability introduced in (Zeng et al., 2015):

**Definition 1** ($\mathcal{C}_\Psi$-ensemble observability). *Suppose $X_t$ follows* (1). *Assume that $\Psi$ is unknown but restricted to a class $\mathcal{C}_\Psi$. We say the tuple $(g, \Xi, \mathcal{C}_\Psi)$ is $\mathcal{C}_\Psi$-ensemble observable if, given $g$, $\Xi$, $\tau$, and all marginals $\mathbf{Q}_t = g_\sharp \mathbf{P}_t$ of $Y_t$ for all $t \in [0, 1]$, the marginals $\mathbf{P}_t$ of $X_t$ are uniquely determined for all $t \in [0, 1]$.*

With this observability assumption, we can infer the latent dynamics solely from the marginals $\mathbf{Q}_t$. A discussion of the relationship between this condition and that of classical/ensemble observability is present in Appendix B. There, we also verify the conditions of Definition 1 for several important setups, e.g., the key velocity-based dynamics model we use in our experiments.

**Theoretical assumptions** We use the following assumptions for the theoretical analysis and provide a more thorough discussion in Appendix C. Let $\mathcal{X}, \mathcal{Y}$ be Polish spaces, where $\mathcal{X}$ is a smooth and compact Riemannian manifold or a compact and convex subset of $\mathbb{R}^d$. In the manifold setting, we assume its Ricci curvature $K$ is bounded from below, e.g. $K > -\infty$.

The path space $\Omega = C([0, 1] : \mathcal{X})$ is equipped with the uniform topology and its Borel $\sigma$-algebra. The probability space on paths $\mathcal{P}(\Omega)$ is equipped with the weak topology, e.g. convergence against bounded, continuous functions. Assume our probability space $(\Omega, \mathcal{F}, \mathbb{P})$ is complete and filtered, where the filtration is with respect to the process $\{X_t\}$. $\mathbb{P}$ is a probability measure, and if it is not specified, expectations are taken with respect to $\mathbb{P}$. Let $\mathbf{W}^{\Xi,\tau}$ be the measure induced by the SDE $dZ_t = -\Xi(t, Z_t)\, dt + \sqrt{\tau}\, dB_t$.

**Assumption 2.1** (Dynamics and Observation Model). *Assume $\Xi : C([0, 1] \times \mathcal{X} : \mathcal{X})$ is known, divergence-free, Lipschitz continuous, and satisfies $\|\Xi\|_{L^\infty} < +\infty$. Assume the observation function $g : \mathcal{X} \to \mathcal{Y}$ is smooth, measurable, bounded, and time invariant.*

The divergence-free assumption is required so that the time marginals of $\mathbf{W}^{\Xi,\tau}$ remain vol for all time if the initial condition is vol,[3] where vol denotes the uniform measure on $\mathcal{X}$. Lipschitz continuity and $\|\Xi\|_{L^\infty} < +\infty$ are technical conditions for our proofs, and there are also some necessary mild technical conditions on the pair $(\Xi, \Psi)$. Finally, by "bounded" $g$ we mean the image of a set of finite measure also has finite measure.

## 3 PO-MFL: Approximate Minimum Entropy Estimation

We seek to estimate the trajectory distribution by maximizing its log-likelihood with respect to the distribution induced by the SDE, subject to matching the observed marginals. Inspired by (Lavenant et al., 2024), we pose the maximum likelihood problem as an equivalent *minimum entropy estimation* problem connecting temporal snapshots into continuous trajectories, where the entropy is the relative entropy (KL divergence) of the estimated trajectory distribution with respect to the known portion of the SDE. We will show that the optimal point of the minimum entropy objective function converges to the ground truth trajectory distribution. It is not practical, however, to directly work with the

---

[3]Note that if $\mathcal{X}$ has a boundary, we need a zero flux condition on $\Xi$.

trajectory distribution as it is an infinite-dimensional object. In what follows, we will show that the minimum entropy objective in trajectory space can be reduced to an OT-based objective, where marginals at adjacent time points are coupled via entropic OT. This reduction allows us to perform trajectory inference using only representations of the latent space time marginals, which can be accomplished via sets of particles. These particles can then be optimized via MFL dynamics.

## 3.1 MINIMUM ENTROPY OBJECTIVE FUNCTION

In this section, we specify the minimum entropy objective function on the trajectory distribution. Let $\{t_i^T\} \subset [0,1]$ be our observation times, where $\Delta t_i := t_{i+1}^T - t_i^T$. Recall that, in general, we do not have exact measurements of the temporal marginals, we will only have samples from them. As a result, we must introduce a *fit function* to measure the discrepancy between the observation space time marginals of the estimated trajectory distribution $\mathbf{R}$ and the observed samples.

Let the observed empirical distribution be $\hat{\rho}_i^T := \frac{1}{N_i^T} \sum_{i=1}^{N_i^T} \delta_{Y_{i,j}^T} \in \mathcal{P}(\mathcal{Y})$ for $i \in [T]$. Consider the fit function $\text{Fit}^{\lambda,\sigma} : \mathcal{P}(\mathcal{Y})^T \to \mathbb{R}$:

$$\text{Fit}^{\lambda,\sigma}(\mathbf{Q}_{t_1^T}, \ldots, \mathbf{Q}_{t_T^T}) := \frac{1}{\lambda} \sum_{i=1}^{T} \Delta t_i \text{DF}^{\sigma}(g_\sharp \mathbf{R}_{t_i^T}, \hat{\rho}_i^T), \tag{3}$$

with data-fitting term introduced by (Chizat et al., 2022) augmented by observation function $g$ to be

$$\text{DF}^{\sigma}(g_\sharp \mathbf{R}_{t_i^T}, \hat{\rho}_i^T) := \int_{\mathcal{Y}} -\log \left[ \int_{\mathcal{X}} \exp \left( -\frac{\|g(x) - y\|^2}{2\sigma^2} \right) d\mathbf{R}_{t_i^T}(x) \right] d\hat{\rho}_i^T(y)$$
$$= H(\hat{\rho}_i^T | g_\sharp \mathbf{R}_{t_i^T} * \mathcal{N}_\sigma) + H(\hat{\rho}_i^T) + C, \tag{4}$$

where $\mathcal{N}_\sigma$ is the Gaussian kernel with variance $\sigma^2$, $C > 0$ is a constant, and we use the substitution $\mathbf{Q}_{t_i^T} = g_\sharp \mathbf{R}_{t_i^T}$. Notice that the inner integral is over $\mathcal{X}$ as the optimization will occur on the latent space, while the inner integral is over $\mathcal{Y}$ as the observations are over $\mathcal{Y}$.

This data-fitting term can be interpreted as the negative log-likelihood under the noisy observation model $\hat{Y}_{i,j}^T = g(X_{i,j}^T) + \sigma Z_{i,j}$, where $\hat{Y}_{i,j}^T$ is the observation and $Z_{i,j} \overset{i.i.d.}{\sim} \mathcal{N}(0, I)$. Note $\text{DF}^{\sigma}$ is jointly convex in $(\mathbf{R}_{t_i^T}, \hat{\rho}_i^T)$ and linear in $\hat{\rho}_i^T$. The main difference compared to (Chizat et al., 2022) is that our data-fitting term is in observation space, e,g. the addition of the function $g$.

The minimum entropy estimator introduced in (Lavenant et al., 2024) minimizes $\mathcal{F} : \mathcal{P}(\Omega) \to \mathbb{R}$

$$\mathcal{F}(\mathbf{R}) := \text{Fit}^{\lambda,\sigma}(\mathbf{Q}_{t_1^T}, \ldots, \mathbf{Q}_{t_T^T}) + \tau H(\mathbf{R}|\mathbf{W}^{\Xi,\tau}). \tag{5}$$

Recall that our key novelties are the fit term in *observation space* and entropy minimization in path space with respect to *divergence-free, Markov* path measures. Nonetheless, we show that we can still recover the ground truth in the limit as the number of observations becomes dense.

**Theorem 3.1** (Consistency (informal, see Thm. C.1)). *Suppose $\mathbf{P}$ is the SDE* (1) *with initial condition $\mathbf{P}_0 \in \mathcal{P}(\mathcal{X})$ such that $H(\mathbf{P}_0|\text{vol}) < +\infty$. Let $\mathbf{R}^{T,\lambda,\sigma} \in \mathcal{P}(\Omega)$ be the unique minimizer of* (5):

$$\mathbf{R}^{T,\lambda,\sigma} := \underset{\mathbf{R} \in \mathcal{P}(\Omega)}{\arg \min} \mathcal{F}(\mathbf{R}).$$

*If $\{t_i^T\}_{t \in [T]}$ becomes dense in $[0,1]$, then $\lim_{\sigma \to 0, \lambda \to 0} \left( \lim_{T \to \infty} \mathbf{R}^{T,\lambda,\sigma} \right) = \mathbf{P}$ almost surely.*

This weak convergence result parallels Theorem 2.3 in (Lavenant et al., 2024), which provides a consistency result in the fully observed setting where the entire state vector $X_t$ is observed and $\Xi = 0$ identically. Due to these differences, the result in (Lavenant et al., 2024) cannot be directly applied to our setting, and while our proof is able to follow a similar overall structure, dense and nontrivial changes throughout the extensive proof are required. See Appendix C.

## 3.2 THE REDUCED PROBLEM

As (5) is an infinite-dimensional optimization problem, to apply the MFL dynamics (Hu et al., 2021), we need to "reduce" the problem over the space $\mathcal{P}(\mathcal{X})^T$, similar to (Chizat et al., 2022). As before,

let $\Delta t_i := t_{i+1}^T - t_i^T$ and $\tau_i := \Delta t_i \cdot \tau$. Consider the following entropic OT cost (see Appendix A), defined for some $\tau_i > 0$, as

$$T_{\tau_i,\Xi}(\mu,\nu) := \min_{\gamma \in \Pi(\mu,\nu)} \int c_{\tau_i}^\Xi(x,y)\, d\gamma(x,y) + \tau_i H(\gamma|\mu \otimes \nu) = \min_{\gamma \in \Pi(\mu,\nu)} \tau_i H(\gamma|p_{\tau_i}^\Xi \mu \otimes \nu), \quad (6)$$

where $p_t^\Xi$ is the transition probability density of $\mathbf{W}^{\Xi,\tau}$ over the time interval $[0,t]$ and the cost function is $c_{\tau_i}^\Xi(x,y) := -\tau_i \log(p_{\tau_i}^\Xi(x,y))$. In general, $p_{\tau_i}^\Xi$ cannot be found explicitly, so we will discuss how to approximate this below. This optimization problem is exactly a Schrödinger bridge problem, e.g. see (Chizat et al., 2022, App. A); (Léonard, 2012; 2013), which inspires the following reduction. Define the functional $G : \mathcal{P}(\mathcal{X})^T \to \mathbb{R}$ for $\boldsymbol{\mu} = (\boldsymbol{\mu}^{(1)}, \ldots, \boldsymbol{\mu}^{(T)})$ that represents a family of reconstructed temporal marginals, by

$$G(\boldsymbol{\mu}) := \mathrm{Fit}^{\lambda,\sigma}(g_\sharp \boldsymbol{\mu}) + \sum_{i=1}^{T-1} \frac{1}{\Delta t_i} T_{\tau_i,\Xi}(\boldsymbol{\mu}^{(i)}, \boldsymbol{\mu}^{(i+1)}), \quad (7)$$

where $g_\sharp \boldsymbol{\mu} \in \mathcal{P}(\mathcal{X})^T$ is an abuse of notation to mean the element-wise push-forward of each $\boldsymbol{\mu}^{(i)}$ by $g$. We consider the *reduced objective* $F : \mathcal{P}(\mathcal{X})^T \to \mathbb{R}$, defined as

$$F(\boldsymbol{\mu}) := G(\boldsymbol{\mu}) + \tau H(\boldsymbol{\mu}), \quad (8)$$

where $H(\boldsymbol{\mu}) = \sum_{i=1}^T \int \log(\boldsymbol{\mu}^{(i)}) d\boldsymbol{\mu}^{(i)}$ is the negative differential entropy of the family of measures $\boldsymbol{\mu}$. Similar to (Chizat et al., 2022), we have an equivalence of minimizing $\mathcal{F}$, (5), the objective in path space $\mathcal{P}(\Omega)$, and $F$, (8), the reduced objective over $\mathcal{P}(\mathcal{X})^T$. The proof is provided in Appendix D.

**Theorem 3.2** (Representer theorem). *Let* $\mathrm{Fit} : \mathcal{P}(\mathcal{Y})^T \to \mathbb{R}$ *be any function and let* $\Xi$ *be bounded and divergence-free.*

(i) *If $\mathcal{F}$ admits a minimizer $\mathbf{R}^*$ then $(\mathbf{R}_{t_1^T}^*, \ldots, \mathbf{R}_{t_T^T}^*)$ is a minimizer for $F$.*

(ii) *If $F$ admits a minimizer $\boldsymbol{\mu}^* \in \mathcal{P}(\mathcal{X})^T$, then a minimizer $\mathbf{R}^*$ for $\mathcal{F}$ is built as*

$$\mathbf{R}^*(\cdot) = \int_{\mathcal{X}^T} \mathbf{W}^{\Xi,\tau}(\cdot|x_1, \ldots, x_T)\, d\mathbf{R}_{t_1^T, \ldots, t_T^T}(x_1, \ldots, x_T),$$

*where $\mathbf{W}^{\Xi,\tau}(\cdot|x_1, \ldots, x_T)$ is the law of $\mathbf{W}^{\Xi,\tau}$ conditioned on passing through $x_1, \ldots, x_T$ at times $t_1^T, \ldots, t_T^T$, respectively and $\mathbf{R}_{t_1^T, \ldots, t_T^T}$ is the composition of the optimal transport plans $\gamma_i$ that minimize $T_{\tau_i,\Xi}(\boldsymbol{\mu}^{*(i)}, \boldsymbol{\mu}^{*(i+1)})$, for $i \in [T-1]$.*

The composition of the transport plans is obtained as:

$$\mathbf{R}_{t_i, \ldots, t_T}(dx_1, \ldots, dx_T) = \gamma_1(dx_1, dx_2)\gamma_2(dx_3|x_2) \cdots \gamma_{T-1}(dx_T|x_{T-1}), \quad (9)$$

where the OT plans $\gamma_i(dx_i, dx_{i+1}) = \gamma_i(dx_{i+1}|x_i)\mu_i(dx_i)$ are conditional probabilities (or "disintegrations"). As in (Chizat et al., 2022), the "reduction" of the optimization space from $\mathcal{P}(\Omega)$ to $\mathcal{P}(\mathcal{X})^T$ is enabled by the Markov property of $\mathbf{W}^{\Xi,\tau}$, which holds for us due to the Lipschitz continuity assumption on $\Xi$ and that $\mathbf{W}^{\Xi,\tau}$ remains the uniform measure at all time. Then, Theorem 3.2 allows us to compute a minimizer for $\mathcal{F}$ from a minimizer for $F$ and its associated OT plans.

## 3.3 Approximating the entropic OT cost

Although now we have a reduced problem over $\mathcal{P}(\mathcal{X})^T$, we are not yet ready to solve the objective function $F$ (8), as the entropic OT cost $T_{\tau_i,\Xi}(\mu,\nu)$ is defined with the transition function $p_{\tau_i}^\Xi$, which is generally not available in closed form. We provide an approximation of $T_{\tau_i,\Xi}(\mu,\nu)$ by considering an Euler-Maruyama discretization (Kloeden & Platen, 1992). Let $\mu \in \mathcal{P}(\Omega)$ be a stochastic process following the SDE $dX_t = -\Xi(t, X_t)\, dt + \sqrt{\tau}\, dB_t$ with an arbitrary initial distribution. Let $\Delta t := t_2 - t_1$ and suppose $\mu_{t_1}, \mu_{t_2}$ are two time marginals of $\mu$. Recall that if $X_{t_1} \sim \mu_{t_1}$, and

$$X_{t_2} := X_{t_1} - \int_{t_1}^{t_2} \Xi(s, X_s)\, ds + \sqrt{\tau}(B_{t_2} - B_{t_1}) \quad (10)$$

then $X_{t_2} \sim \mu_{t_2}$. For small $\Delta t = t_2 - t_1$, since $\Xi$ and $\mu_s$ are smooth, the integrand of the second term will be approximately constant over the integration interval. Thus, we can approximate the first two terms of (10) as

$$\xi^{\Delta t}(X_{t_1}) := X_{t_1} - \Xi(t_1, X_{t_1}) \cdot \Delta t.$$

Finally, the last term of (10) is an isotropic Gaussian with variance $\tau \Delta t$. This suggests approximating the transition kernel $p_{\tau_i}^{\Xi}$ as a deterministic drift given by the current $\Xi$, followed by isotropic Gaussian noise.[4] For $\Xi = 0$, this would reduce to the kernel used by the baseline method, i.e. the Brownian motion transition kernel.

This provides an intuition for why our approach is more robust than that of the baseline method when the true $\Xi$ is non-zero, since $\xi^{\Delta t}(X_{t_1}) - X_{t_2} \approx \mathcal{N}(0, \tau \Delta t)$ for small $\Delta t$, while the $X_{t_1} - X_{t_2}$ used by the baseline algorithm has non-zero expectation, is non-Gaussian, and often has significantly higher variance. In a sense, our Euler-Maruyama approximation can be considered a *first order* approximation method, while the baseline corresponds to a *zeroth order* method.

To formalize this intuition, first let $\mathrm{TV}(\cdot, \cdot)$ denote the total variation distance. We have the following bound on the densities, which is a special case of (Bras et al., 2022, Thm. 2.1). Note that this implies that the difference in probability between the approximate kernel and true kernel for any event is of order $O((\Delta t)^{1/3})$, which converges to 0 as $T \to \infty$.

**Proposition 3.3.** *Let $X_{t_2}$ be as in (10) and $\tilde{X}_{t_2} := \xi^{\Delta t}(X_{t_1}) + \sqrt{\tau}(B_{t_2} - B_{t_1})$, where $X_{t_1} = \delta_x$. Then,* $\mathrm{TV}(X_{t_2}, \tilde{X}_{t_2}) \le C_1 e^{C_2 |x|^2} (\Delta t)^{1/3}$, *where the constants $C_1, C_2 > 0$ depend only on $\dim \mathcal{X}$ and the Lipschitz constant of $\Xi$.*

Applying this approximate transition kernel yields updated, tractable OT terms in the objective function. Specifically, instead of $T_{\tau_i, \Xi}(\boldsymbol{\mu}^{(i)}, \boldsymbol{\mu}^{(i+1)})$ in (7), we use $T_{\tau_i}(\xi_{\sharp}^{t_{i+1}-t_i} \boldsymbol{\mu}^{(i)}, \boldsymbol{\mu}^{(i+1)})$, where

$$T_{\tau_i}(\mu, \nu) := \min_{\gamma \in \Pi(\mu, \nu)} \tau_i H(\gamma | p_{\tau_i} \mu \otimes \nu)$$

and $p_t(x, y)$ is the transition probability density of the Brownian motion on $\mathcal{X}$ over the time interval $[0, t]$. This cost is easily computed as $p_{\tau_i}$ is the Gaussian kernel. In particular the cost function is $\tilde{c}_{\tau_i}^{\Xi}(x, y) := -\tau_i \log(p_{\tau_i}(\xi^{\Delta t}(x), y))$, and we use Varadhan's approximation

$$\tilde{c}_{\tau_i}^{\Xi}(x, y) \approx \frac{1}{2} \|y - x + \Delta t \cdot \Xi(t_1, x)\|^2,$$

which holds for $\tau_i$ small (Norris, 1997). To wrap up our discussion here, it is important to highlight that we require a generalization of (Lavenant et al., 2024, Thm. 2.3) using our path measure $\mathbf{W}^{\Xi, \tau}$, Theorem 3.1, to justify convergence of our estimator when including $\Xi$ in our cost function in the entropic OT problem (6).

## 3.4 MEAN-FIELD LANGEVIN DYNAMICS AND EXPONENTIAL CONVERGENCE

We provide a brief description of the MFL dynamics. See Appendix E for a more complete discussion. MFL dynamics are designed to minimize functionals of the form $F_\tau = G + \tau H$, where $G : \mathcal{P}(\mathcal{X}) \to \mathbb{R}$ is smooth and $H$ is minus the differential entropy. Using the first-variation $V[\boldsymbol{\mu}]$ of $G$ given in Proposition E.1, the MFL dynamics is defined as the solution of the following non-linear McKean-Vlasov SDE, for $s \ge 0$:

$$\begin{cases} dX_s^{(i)} = -\nabla V^{(i)}[\boldsymbol{\mu}_s](X_s^{(i)}) \, ds + \sqrt{2\tau} \, dB_s^{(i)} + d\Phi_s^{(i)}, & \mathrm{Law}(X_0^{(i)}) = \boldsymbol{\mu}_0^{(i)} \\ \boldsymbol{\mu}_s^{(i)} = \mathrm{Law}(X_s^{(i)}), & i \in [T], \end{cases} \quad (11)$$

where $d\Phi_s^{(i)}$ is the boundary reflection in the sense of the Skorokhod problem, e.g. Tanaka (1979). Note that $\Xi$ affects this equation implicitly via the Schrödinger potentials. For computation, we discretize the MFL dynamics via noisy particle gradient descent, discussed in Appendix E.2.

In (Chizat et al., 2022; Chizat, 2022; Nitanda et al., 2022), it is shown that the MFL dynamics converges at an exponential rate to the minimizer. We provide the proof for the following similar result for our partially observed setting in Appendix E.[5]

---

[4] Alternative discretizations can be considered, e.g. we use a midpoint discretization for the constant-velocity experiments, but for clarity, we introduce the method using a standard Euler discretization.

[5] The $d$-torus assumption is required for technical reasons in (Chizat, 2022).

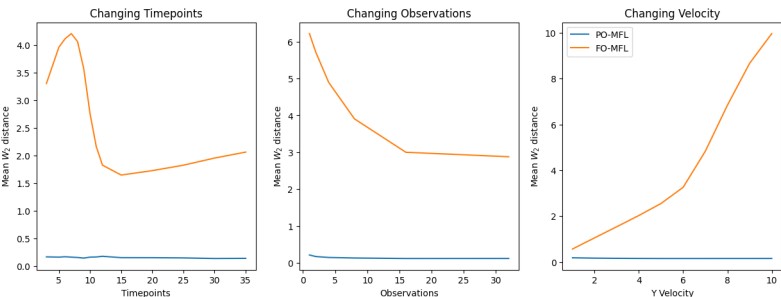

Figure 3: Average $W_2$ distance between ground truth and PO-MFL (blue) and FO-MFL (orange) recovered positions in "constant velocity" model (App. F). (L) Number of time points. (C) Number of observations. (R) Velocity.

**Theorem 3.4** (Convergence). *Assume $\mathcal{X}$ is the d-torus. Let $\boldsymbol{\mu}_0 \in \mathcal{P}(\mathcal{X})^T$ be such that $F(\boldsymbol{\mu}_0) < +\infty$. Then for $\tau \geq 0$, there exists a unique solution $(\boldsymbol{\mu}_s)_{s \geq 0}$ to the MFL dynamics (11). Let $\tau > 0$ and assume that $\boldsymbol{\mu}_0$ has a bounded absolute log-density, it holds*

$$F_\tau(\boldsymbol{\mu}_s) - \min F_\tau \leq e^{-Cs}(F_\tau(\boldsymbol{\mu}_0) - \min F_\tau),$$

*where $C = \beta e^{-\alpha/\tau}$ for some $\alpha, \beta > 0$ independently of $\boldsymbol{\mu}$ and $\tau$. Moreover, taking a smooth time-dependent $\tau_s$ that decays asymptotically as $\tilde{\alpha}/\log s$ for some $\tilde{\alpha} > \alpha$, it holds $F_0(\boldsymbol{\mu}_s) - F_0(\boldsymbol{\mu}^*) \lesssim \log\log s/\log s \to 0$ and $\boldsymbol{\mu}_s$ converges weakly to the min-entropy estimator $\boldsymbol{\mu}^*$.*

## 3.5 PO-MFL

We summarize our proposed latent trajectory inference method in Algorithm 1. We recall that discussion on the cost function (line 4) and MFL dynamics (line 7) can be found in Sections 3.3 and 3.4, respectively. We use the Sinkhorn algorithm for entropic OT, which we discuss in Appendix A. Using $N$ iterations of the MFL dynamics, the total runtime for our algorithm is $O(NTm^2)$, as we need to solve $T-1$ entropic OT problems on $m \times m$ size matrices in each iteration, where we assume that the number of iterations in the Sinkhorn algorithm is capped at a constant. We remark that our PO-MFL method has the same runtime as the baseline FO-MFL (fully-observed MFL).

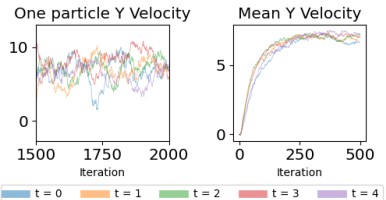

Figure 2: (left) Velocity of one particle at end of optimization. (right) Population velocity at beginning of optimization, showing exponential convergence.

---

**Algorithm 1** PO-MFL: framework for latent trajectory inference

---

**Require:** Collection of observations $(\hat{\rho}_1, \ldots, \hat{\rho}_T)$, collection of $T$ time samples $(t_1^T, \ldots, t_T^T)$, velocity dynamics $\Xi$, number of iterations for MFL dynamics $N$, number of particles $m$, entropic OT parameter $\lambda$
1: Initialize $m$ particles for each time: $(\hat{m}_1, \ldots, \hat{m}_T) \in \mathcal{X}^{m \times T}$
2: **for** $N$ iterations **do**
3:     **for** $i \in [T-1]$ **do**
4:         $\Delta t_i := t_{i+1}^T - t_i^T$
5:         $C_i := \{C_{j,k}\}_{j,k=1}^m \leftarrow \frac{1}{2}\|\hat{m}_{i+1,k} - \hat{m}_{i,j} + \Delta t_i \Xi(t_i^T, \hat{m}_{i,j})\|^2$
6:         $\gamma_i \leftarrow \text{Sinkhorn}(\hat{m}_i, \hat{m}_{i+1}, C_i, \lambda \cdot \Delta t_i)$
7:     **end for**
8:     $\hat{\mathbf{m}} \leftarrow \text{MFL}(\hat{\mathbf{m}}, \boldsymbol{\gamma}, \hat{\boldsymbol{\rho}})$                    ▷ $\hat{\mathbf{m}} := (\hat{m}_1, \ldots, \hat{m}_t)$, etc.
9: **end for**
10: Output collection of particles $\hat{\mathbf{m}}$, trajectories $\gamma_{t-1} \circ \cdots \circ \gamma_1$

---

Given the set of observed temporal marginal samples in the *observation* space $\mathcal{Y}$, Algorithm 1 yields a set of $m$ particles at each time step $i$ representing the temporal marginal distributions in the *latent* space $\mathcal{X}$. Simulated trajectories may be recovered by sampling from the composition of entropic transport plans as shown in (9).

## 4 EXPERIMENTS

In this section, we briefly provide synthetic experiments that demonstrate the advantages of having a dynamics prior. More details on the experimental setup can be found in Appendix F.[6]

**"Constant velocity" model** We compare the behavior of our method, PO-MFL, to that of the baseline, FO-MFL, using the "constant velocity" model popular in target tracking, e.g. (McIntyre & Hintz, 1998). We provide further details of this model and its ensemble observability in Appendix B.1. This model can be interpreted as introducing velocity as a hidden state to be inferred, in order to build *momentum* into the dynamics (an object in motion tends to stay in motion). This is an extremely generic model and makes minimal assumptions on the underlying data, as evidenced by its use in target tracking.

In this model, the state space is $X = (x, y, \dot{x}, \dot{y}) \in \mathbb{R}^4$, with $\Xi$ given in the appendix and observations $g(X) = [I_2, 0_{2 \times 2}]X$. Note that due to non-zero process noise $\tau$, despite the name, this model does not imply that the velocity is constant in time. The particles are initialized at the origin with velocities set as $\dot{x} = 5$ and $\dot{y} = 7$, i.e. $X_0 = (0, 0, 5, 7)$. The ground truth is shown in Figure 1a.

Our optimization method observes only the positions of the particles, i.e. $g(x, y, \cdot, \cdot) = (x, y)$, but it uses $\Xi$ as being a constant velocity prior. Results shown

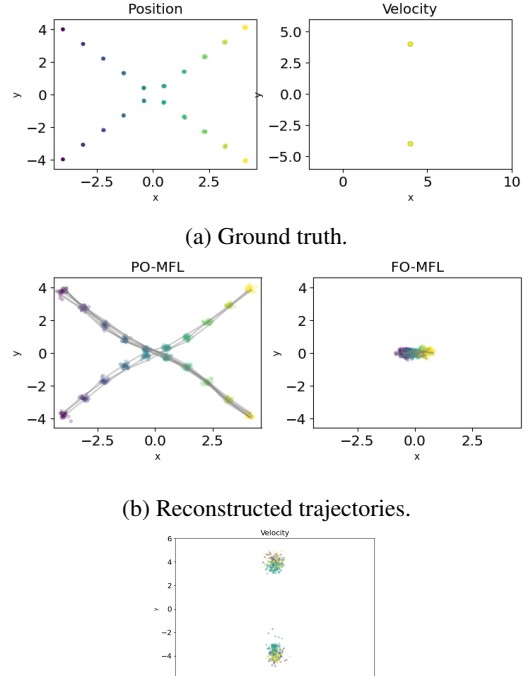

(a) Ground truth.

(b) Reconstructed trajectories.

(c) Reconstructed velocity from PO-MFL. Note the bimodal velocity estimate.

Figure 4: Crossing paths experiment under the "constant velocity" SDE.

in Figure 1b show that PO-MFL is able to successfully reconstruct the paths trajectories, while FO-MFL fails to converge. Furthermore, in Figure 1c, we verify that the population average of the particles' velocities matches with the ground truth.

Figure 2 (left) displays the $y$ velocity of one particle for the last 500 iterations of optimization. Although at each iteration, the velocity is stochastic, we can see that the mean is at 7. In Figure 2 (right), we plot the average $y$ velocity in the first 400 iterations of optimization, providing empirical evidence of the exponential convergence of our algorithm guaranteed by Theorem 3.4.

Figure 4 shows a crossing paths experiment where the population is divided into two groups, one moving right and down, and the other right and up, with their paths crossing in the middle. In this particularly illuminating regime, PO-MFL leverages the "constant velocity" model used in this section to distinguish the downward moving group from the upward moving group. Note that PO-MFL is not told a priori which samples belong to which group. While FO-MFL here collapses to the centroid, we point out that even if its optimization was successful, FO-MFL would prefer $V$-shaped trajectories, e.g. those that have velocity $v$ for half the time and then $-v$ for the remainder, here rather than the correct straight-line trajectories, as it does not retain a hidden velocity state and only seeks to match adjacent time points by their relative position via entropic OT.

We provide a variety of additional experiments in Appendix F illustrating how performance changes as the number of observed particles, the spacing of time points, and the underlying ground truth initial velocity affects performance. Figure 3 shows the average $W_2$ distance between the ground truth positions and recovered positions (averaged by time point) across these experiments. Our approach remains significantly more robust as these parameters are varied compared to FO-MFL.

---

[6]We provide code in https://github.com/AnmingGu/partially-observed-traj-inference.

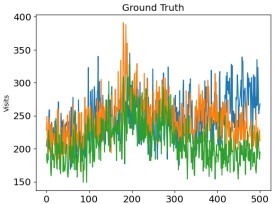
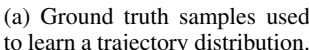
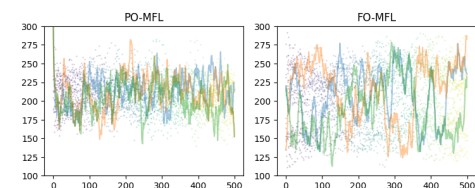

(a) Ground truth samples used to learn a trajectory distribution.

(b) Samples from the learned trajectory distribution (lines), overlaid with scatter plots of the learned particle representations of the time marginals.

Figure 5: Wikipedia page traffic data.

**Wikipedia traffic data** We briefly consider real-world daily traffic data from Wikipedia.[7] Because some of the webpage traffic has large spikes/outliers, we only consider pages whose daily visits are between 100 and 500. We use the data from 3 pages over the course of 500 days as our ground truth and seek to learn the trajectory distribution.

For PO-MFL, we do not consider partial observations. Instead, we leverage the dynamics prior $\Xi$ to set an autoregressive model $x_{t+1} = \theta_1 x_t + \theta_2 x_{t-6}$ and let $g$ be the identity function. In this setting, the state space for FO-MFL is $\mathbb{R}^{500}$, while the state space for PO-MFL is $\mathbb{R}^{2 \times 500}$. The first marginal of PO-MFL matches the data for FO-MFL, while the second marginal of PO-MFL is the data lagged by 6 days. For the values of $\theta_1$ and $\theta_2$, we take the average of the regression parameters calculated from 30 trajectories drawn from the same distribution.

We try different number of particles $m = 3, 6, 15$ and report the results in Table 1. To compute the values in the table we sample 3 trajectories from the output of the algorithm and use this as our empirical measure. We use the Wasserstein distance in $\mathbb{R}^{500}$ as our cost metric. Here, we use just the first dimension for PO-MFL. We see that both the mean and variance from PO-MFL are significantly less than that of FO-MFL. We plot three trajectories in Figure 5 with the $m = 3$ experiment, and note that PO-MFL yields less volatile trajectories more in line with the ground truth trajectories.

| $m$ | PO-MFL | FO-MFL |
|---|---|---|
| 3 | **1.01±0.0395** | 1.61±0.206 |
| 6 | **1.07±0.0726** | 1.76±0.303 |
| 15 | **1.05±0.0956** | 1.91±0.443 |

Table 1: Average (normalized) $W_2$ distance between true and sampled trajectory distributions for Wikipedia dataset over 100 MCMC trials. Error bars are one standard deviation.

## 5 CONCLUSION

We consider the problem of trajectory inference in latent space based on indirect observation, extending the theoretical analysis of the min-entropy estimator introduced in (Lavenant et al., 2024) and the MFL dynamics algorithm introduced in (Chizat et al., 2022). Experiments were provided showing that the ability to include simple non-informative latent dynamics models, such as the "constant velocity" model, and autoregressive models, can dramatically improve the trajectory inference performance over the baseline FO-MFL method. For future work, while we do here provide some flexibility for model misspecification via the unknown $\Psi$ potential, it would be interesting to further explore the stability of our method when the dynamics model $\Xi$ is misspecified. Further exploration of ensemble observability to stochastic systems would also be a highly interesting fundamental direction to explore. Finally, we will seek to explore the various promising empirical use cases outlined in the introduction.

## ACKNOWLEDGEMENTS

The authors would like to thank Hugo Lavenant for helpful discussions and the anonymous reviewers for their feedback.

---

[7] We use `train_2.csv` from https://www.kaggle.com/competitions/web-traffic-time-series-forecasting

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

## A ENTROPIC OPTIMAL TRANSPORT

We provide a brief exposition to entropic OT. We refer the reader to (Cuturi, 2013; Peyré & Cuturi, 2020) for a more thorough introduction.

Let $\mathcal{X}, \mathcal{Y}$ be arbitrary spaces, $c : \mathcal{X} \times \mathcal{Y} \to \mathbb{R}$ be a cost function, and $\mu, \nu$ be probability measures on $\mathcal{X}, \mathcal{Y}$, respectively. The entropic OT problem is

$$T_\epsilon(\mu, \nu) = \inf_{\pi \in \Pi(\mu, \nu)} \int_{\mathcal{X} \times \mathcal{Y}} c(x, y)\pi(dx, dy) + \epsilon H(\pi | \mu \otimes \nu),$$

where $\Pi(\mu, \nu)$ is the set of all probability measures on $\mathcal{X} \times \mathcal{Y}$ with marginals $\mu$ on $\mathcal{X}$ and $\nu$ on $\mathcal{Y}$, $H$ is the relative entropy, and $\epsilon$ is the regularization parameter. By standard duality theory, this is equivalent to the following problem

$$T_\epsilon(\mu, \nu) = \max_{\varphi \in L^1(\mu), \psi \in L^1(\nu)} \int \varphi d\mu + \int \psi d\nu + \epsilon \left( 1 - \int e^{\frac{1}{\epsilon}(\varphi(x) + \psi(y) - c(x,y))} d\mu(x) d\mu(y) \right),$$

which admits a unique solution up to a translation $(\varphi + \kappa, \psi - \kappa)$ for $\kappa \in \mathbb{R}$. Furthermore, the functions $(\varphi, \psi)$ satisfy the following conditions:

$$\begin{cases} \varphi(x) = -\epsilon \log \int \exp(\frac{1}{\epsilon}(\varphi(y) - c(x,y))) d\nu(y) \\ \psi(y) = -\epsilon \log \int \exp(\frac{1}{\epsilon}(\varphi(y) - c(x,y))) d\mu(x). \end{cases}$$

In the discrete (empirical measure) setting, these potentials give motivation for the Sinkhorn algorithm, which we describe in Algorithm 2. Here, $\odot$ is taken to be element-wise multiplication. (Altschuler et al., 2017) shows that the entropic OT problem can be solved in approximately linear time.

---

**Algorithm 2** Sinkhorn

---

**Require:** Probability measures $\mu, \nu$, cost matrix $C$, regularization parameter $\epsilon$, number of iterations $N$
1: $\varphi^{(0)} \leftarrow \mathbf{1}$
2: $K \leftarrow \exp(-C/\epsilon)$
3: **for** $i = 1, \ldots, N$ **do**
4: $\quad \psi^{(i)} \leftarrow \nu \odot K^\top \varphi^{(i-1)}$
5: $\quad \varphi^{(i)} \leftarrow \mu \odot K \psi^{(i-1)}$
6: **end for**
7: Output transport plan $\operatorname{diag}(\varphi^{(N)}) K \operatorname{diag}(\psi^{(N)})$

---

## B ENSEMBLE OBSERVABILITY FOR LINEAR SYSTEMS

Recall that in classical observability (Gajic et al., 1996), the goal is to recover the dynamics of a single particle, while here we want to recover the dynamics of a probability distribution. The notion of ensemble observability introduced in (Zeng et al., 2015) tackles this problem. Consider the non-stochastic model with linear $\Xi(X) = A_\Xi X + B_\Xi$:

$$dX_t = -(A_\Xi X_t + B_\Xi)\, dt \tag{12}$$

with initial condition $\mathbf{P}_0$ and linear observations $Y_t = g(X_t) = C_g X_t + D_g$. For shorthand, we denote this system as $(A_\Xi, B_\Xi, C_g, D_g)$.

The following is the definition of ensemble observability as introduced in (Zeng et al., 2015): it does not consider stochasticity.

**Definition 2** (Ensemble observability (Zeng et al., 2015, Def. 1)). *The linear system* (12) *is* ensemble observable *if given marginals $g_\sharp \mathbf{P}_t$ of $Y_t$ for all $t \in [0, 1]$, the marginals $\mathbf{P}_t$ of $X_t$ are uniquely determined for all $t \in [0, 1]$.*

We can consider Definition 1 as an extension of ensemble observability. In particular, if we let $\tau = 0$ in Definition 1, we exactly recover ensemble observability. (Zeng et al., 2015) showed that classical observability is a necessary condition for ensemble observability, and provided several sufficient conditions as well. For a random variable $X$, we denote $\varphi_X$ to be its characteristic function. We assume the following on the initial distribution $X_0$.

**Assumption B.1.** *Let $X_0$ be such that $s \mapsto \varphi_{X_0}(sv)$ is real-analytic for all non-zero $v \in \mathbb{R}^n$.*

This assumption is not very strong and most "nice" distributions satisfy it, e.g. if they admit have a density with respect to the Lebesgue measure. Recall that by (Gajic et al., 1996, Thm. 5.2), classical observability holds if and only if the observability matrix $O := [C_\Xi, C_\Xi A_\Xi, \ldots, C_\Xi A_\Xi^{n-1}]$ has rank $n$. (Zeng et al., 2015) provides two useful sufficient conditions[8] for ensemble observability of such systems.

**Proposition B.2** ((Zeng et al., 2015, Thm. 8)). *Under Assumption B.1, if $(A_\Xi, B_\Xi, C_g, D_g)$ is observable and $\operatorname{rank} C_g = n - 1$, then $(A_\Xi, B_\Xi, C_g, D_g)$ is ensemble observable.*

**Corollary B.3** ((Zeng et al., 2015, Cor. 8)). *Under Assumption B.1, if $\mathcal{X} = \mathbb{R}^2$ and $(A_\Xi, B_\Xi, C_g, D_g)$ is observable, then $(A_\Xi, B_\Xi, C_g, D_g)$ is ensemble observable.*

We now show that these two conditions can be carried over to stochastic systems, i.e., those with $\tau > 0$. Consider adding stochasticity to (12) with the model

$$dX_t = -(A_\Xi X_t + B_\Xi)\, dt + \sqrt{\tau}\, dB_t \tag{13}$$

with initial condition $\mathbf{P}_0$, $\{B_t\}$ is an $\mathbb{R}^n$-valued Brownian motion, and observations $Y_t = C_g X_t + D_g$. We have the following result:

**Corollary B.4.** *Suppose $(A_\Xi, B_\Xi, C_g, D_g)$ is ensemble observable (with $\tau = 0$). Then the system (13) with known $\tau > 0$ is ensemble observable.*

*Proof.* It is easy to see via direct calculation, e.g. using an integrating factor, that the solution to (13) is

$$X_t = e^{-A_\Xi t} X_0 - \left( \int_0^t e^{-A_\Xi(t-s)}\, ds \right) B_\Xi + \sqrt{\tau} \int_0^t e^{-A_\Xi(t-s)}\, dB_s, \tag{14}$$

where we use matrix exponentials. Using arguments similar to those in (Vatiwutipong & Phewchean, 2019), we can characterize the covariance:

$$\Sigma_t := \operatorname{Cov}[X_t] = \tau \int_0^t e^{-(A_\Xi + A_\Xi^\top)(t-s)} ds.$$

We also know that

$$\mu_t := \mathbb{E}[X_t] = e^{-A_\Xi t} X_0 - \left( \int_0^t e^{-A_\Xi(t-s)}\, ds \right) B_\Xi.$$

Then as the first two terms on the right-hand side of (14) have zero variance and the Itô integral of a deterministic integrand is normally distributed with mean zero, we know that (14) is distributed as

$$X_t \sim \mathcal{N}(\mu_t, \Sigma_t).$$

As we know $\tau$, and the corresponding observability matrix to the system has full-rank, we see that the pushforward (to observation space) of every term in (14) is also fully recoverable as well. Note that it is possible to deconvolve, e.g., using the fact that deconvolution is equivalent to division in the Fourier domain, the known Gaussian noise from $X_t$, and hence the system is ensemble observable. This concludes the proof. $\square$

Next, we extend Corollary B.3 to independent processes and apply Corollary B.4.

**Proposition B.5.** *Let $\mathcal{X} = \mathcal{X}_1 \times \cdots \times \mathcal{X}_n$, where each $\mathcal{X}_i = \mathbb{R}^2$. Suppose $(A_{\Xi_i}, B_{\Xi_i}, C_{g_i}, D_{g_i})$ is observable for each $i \in [n]$. Further, suppose the initial condition joint distribution for the $X_{0,i}$ satisfies Assumption B.1 with $[X_{0,i}]_1, [X_{0,j}]_2$ conditionally independent conditioned on $[X_{0,i}]_2, [X_{0,j}]_1$, for each $j \neq i$. If noise parameter $\tau > 0$ is known and $\{B_t\}$ is an $\mathcal{X}$-valued Brownian motion, then the system*

$$\left( \begin{pmatrix} A_{\Xi_1} \\ \vdots \\ A_{\Xi_n} \end{pmatrix}, \begin{pmatrix} B_{\Xi_1} \\ \vdots \\ B_{\Xi_n} \end{pmatrix}, \begin{pmatrix} C_{g_1} \\ \vdots \\ C_{g_n} \end{pmatrix}, \begin{pmatrix} D_{g_1} \\ \vdots \\ D_{g_n} \end{pmatrix} \right)$$

---

[8]These are not the only concrete conditions provided therein. Furthermore, a more general sufficient condition is provided which is possible to check numerically.

*is ensemble observable. Furthermore, the system remains ensemble observable under (known) permutations.*

*Proof.* This follows from a simple application of Corollaries B.3 and B.4. □

### B.1 EXAMPLE: "CONSTANT VELOCITY" MODEL

The two-dimensional "constant velocity" model, so named because the velocity would be constant if there were no process noise ($\tau = 0$), uses a state vector

$$X = (x, y, \dot{x}, \dot{y}) \in \mathbb{R}^4,$$

where here $(x, y)$ are two-dimensional positional coordinates, and $(\dot{x}, \dot{y})$ is the current two-dimensional velocity. The "constant velocity" dynamics model uses $\Xi(X) = A_\Xi X$ where

$$A_\Xi = \begin{bmatrix} 0 & 0 & 1 & 0 \\ 0 & 0 & 0 & 1 \\ 0 & 0 & 0 & 0 \\ 0 & 0 & 0 & 0 \end{bmatrix},$$

which is a very simple matrix simply implying that the rate of change of $X_1 = x$ is given by the current state vector $X_3 = \dot{x}$, and similarly for $y$.

Having defined the dynamics, in this model, only the positions are observed. In other words, the observations $g(X) = [I_2, 0_{2\times 2}]X$, i.e. $C_\Xi = [I_2, 0_{2\times 2}]$ and $D_\Xi = 0$.

Note that this experimental setting satisfies Proposition B.5 as the $x$ and $y$ dynamics are independent. Hence, it is sufficient to check if the system is observable. Here $C = [1, 0]$ and $A = [0, 1; 0, 0]$, so the observability matrix for each of these subsystems becomes

$$\begin{bmatrix} C \\ CA \end{bmatrix} = \begin{bmatrix} 1 & 0 \\ 0 & 1 \end{bmatrix},$$

which is the identity and thus full rank. By observability theory, the system is classically observable, and by the results above, ensemble observable as well.

Note that ensemble observability can be extended to non-zero $\Psi$ in this setting. For instance, in the "constant velocity" model of the main text, $\nabla\Psi = [0; \psi]$ for constant but unknown $\psi$ will serve simply as a drift term on the mean of the hidden velocity state. Since without this drift the mean of the velocity is constant, this drift will be identifiable and the system will be ensemble observable.

Finally, as a brief remark, note that in Definition 1, we require $\Psi$ to be restricted to a class of functions $\mathcal{C}_\Psi$ as otherwise the SDE (1) may fail to satisfy classical observability. Further exploration of $\mathcal{C}_\Psi$ classes is left to future work.

## C PROOFS FOR CONSISTENCY

We make this section as self-contained as possible, although we suppress some of the longer details when they are very similar to certain corresponding results in (Lavenant et al., 2024). Whenever we do, we point to the fuller arguments in (Lavenant et al., 2024). We will use $\Phi_\sigma$ to denote the $\sigma$-wide heat kernel, which is equivalent to the Gaussian convolution $\mathcal{N}_\sigma$. The main result in this section is the following:

**Theorem C.1** (Consistency, (formal version of Thm. 3.1)). *Let $\mathbf{P}$ be the law of the SDE given in* (1)*, restated below:*

$$dX_t = -\Xi(t, X_t)\,dt - \nabla\Psi(t, X_t)\,dt + \sqrt{\tau}\,dB_t,$$

*with initial condition $\mathbf{P}_0 \in \mathcal{P}(\mathcal{X})$ such that $H(\mathbf{P}_0|\mathrm{vol}) < +\infty$. Assume we have the following:*

*(i) $g : \mathcal{X} \to \mathcal{Y}$ is a smooth, measurable, bounded, time invariant function, and $(g, \Xi, \mathcal{C}_\Psi)$ is $\mathcal{C}_\Psi$-ensemble observable.*

*(ii) For every $T \geq 1$, we have a sequence of ordered observation times $\{t_i^T\}_{i=1}^T$ between 0 and 1, and $\{t_i^T\}_{i=1}^T$ becomes dense in $[0, 1]$ as $T \to +\infty$.*

(iii) *For each $T$ and each $i \in [T]$, we have $N_i^T \geq 1$ random variables $\{Y_{i,j}^T\}_{j=1}^{N_i^T}$, which are i.i.d. and distributed according to $g_\sharp \mathbf{P}_{t_i^T}$.*

(iv) *The variables $Y_{i,j}^T$ and $Y_{i',j'}^{T'}$ are sampled independently from their respective distributions except when $(T, i, j) = (T', i', j')$.*

*Consider the functional* (5)*, restated below:*

$$\mathcal{F}(\mathbf{R}) := \mathrm{Fit}^{\lambda,\sigma}(g_\sharp \mathbf{R}_{t_1^T}, \ldots, g_\sharp \mathbf{R}_{t_T^T}) + \tau H(\mathbf{R}|\mathbf{W}^{\Xi,\tau}),$$

*and let $\mathbf{R}^{T,\lambda,\sigma} \in \mathcal{P}(\Omega)$ be its unique minimizer:*

$$\mathbf{R}^{T,\lambda,\sigma} := \underset{\mathbf{R} \in \mathcal{P}(\Omega)}{\arg\min} \mathcal{F}(\mathbf{R}).$$

*Then, we have the weak convergence*

$$\lim_{\sigma \to 0, \lambda \to 0} \left( \lim_{T \to \infty} \mathbf{R}^{T,\lambda,\sigma} \right) = \mathbf{P},$$

*almost surely.*

*Proof.* We use Theorem C.5 to take the limit $T \to +\infty$. By the law of large numbers and the weak convergence assumption, we have $\overline{\rho}_t = \mathbf{P}_t$, almost surely. Define $\mathbf{R}^{\lambda,\sigma}$ to be the limit of $\mathbf{R}^{T,\lambda,\sigma}$ as $T \to +\infty$. By Theorem C.5, it is the unique minimizer of

$$\mathbf{R} \mapsto F_{\lambda,\sigma}(\mathbf{R}) := \frac{1}{\lambda} \int_0^1 \mathrm{DF}(g_\sharp \mathbf{R}_t, \mathbf{P}_t) \, dt + \tau H(\mathbf{R}|\mathbf{W}^{\Xi,\tau}).$$

By definition of the data-fitting term, the functional $G_{\lambda,\sigma}$ in Theorem C.17 differs from $F_{\lambda,\sigma}$ only by a constant. We see that

$$G_{\lambda,\sigma}(\mathbf{R}) = F_{\lambda,\sigma}(\mathbf{R}) - \int_0^1 H(\mathbf{P}_t) \, dt - C,$$

so $\mathbf{R}^{\lambda,\sigma}$ must also be the unique minimizer for $G_{\lambda,\sigma}$. Finally, we use Theorem C.17 to take the limit of $\mathbf{R}^{\lambda,\sigma}$ as $\sigma \to 0$ and $\lambda \to 0$. This concludes the proof. $\qquad\square$

## C.1 VARIATIONAL CHARACTERIZATION OF THE SDE

We recall some previously introduced preliminaries and notation. $\Omega = C([0, 1] : \mathcal{X})$ is the set of $\mathcal{X}$-valued paths, $\{X_t\}_{t \in [0,1]}$ is our canonical process, and $\mathcal{F}$ is the Borel $\sigma$-algebra generated by the random variables $X_s$ for $s \leq t$ such that $\{\mathcal{F}_t\}_{t \in [0,1]}$ is a filtration. We use the notation $|\cdot|^2 = \langle \cdot, \cdot \rangle \in \mathbb{R}$ for the quadratic variation of a process (similarly use this notation for cross-variation).

For the Girsanov transforms to be martingales, we have the following mild technical assumption.

**Assumption C.2** (Novikov conditions on $\Xi$). *Assume that the following Novikov conditions hold:*

$$\mathbb{E}\left[ \exp\left( \frac{1}{2} \int_0^1 |\Xi|^2 \, ds \right) \right] < +\infty$$

*and*

$$\mathbb{E}\left[ \exp\left( \frac{1}{2} \int_0^1 |\Xi + \nabla\Psi|^2 \, ds \right) \right] < +\infty.$$

*Also, assume that there exists $C < +\infty$ such that $\int_0^1 |\Xi|^2 \, ds \leq C$ and $\int_0^1 |\Xi + \nabla\Psi|^2 \, ds \leq C$.*

This last condition is so that we can apply Girsanov's on manifolds, e.g. (Hsu, 2002, Thm. 8.1.2).

**Proposition C.3** (analogous to (Lavenant et al., 2024, Prop. 2.11)). *Let $\mathbf{P}$ be the law of the SDE in* (1)*. Then the Radon-Nikodym derivative of $\mathbf{P}$ with respect to $\mathbf{W}^{\Xi,\tau}$ is given $\mathbf{W}^{\Xi,\tau}$-a.e. by*

$$\frac{d\mathbf{P}}{d\mathbf{W}^{\Xi,\tau}}(X) = \frac{d\mathbf{P}_0}{d\mathrm{vol}}(X_0) \exp\left( \frac{\Psi(0, X_0) - \Psi(1, X_1)}{\tau} \right)$$

$$\cdot \exp\left( \frac{1}{\tau} \left( \int_0^1 \left( \partial_s \Psi - \frac{1}{2}|\nabla\Psi|^2 - \langle \Xi, \nabla\Psi \rangle + \frac{\tau}{2}\Delta\Psi \right) (s, X_s) \, ds \right) \right). \tag{15}$$

To prove this proposition, we do not use a martingale characterization as in (Lavenant et al., 2024), but directly use the Girsanov theorem (which by our assumption on $\Xi$, can be applied on manifolds) and the Itô formula.

*Proof.* By the chain rule, we have

$$\frac{d\mathbf{P}}{d\mathbf{W}^{\Xi,\tau}}(X) = \frac{d\mathbf{P}_0}{d\text{vol}}(X_0) \cdot \frac{d\mathbf{P}}{d\mathbf{W}^\tau} \cdot \frac{d\mathbf{W}^\tau}{d\mathbf{W}^{\Xi,\tau}}.$$

The first term follows from an averaging argument identical to that of (Lavenant et al., 2024, Prop. 2.11). For the second term, recall that $\mathbf{P}$ is the measure induced by the process $dX_t = -(\Xi + \nabla\Psi)\,dt + \sqrt{\tau}\,dB_t$ and $\mathbf{W}^\tau$ is the measure induced by the process $dY_t = \sqrt{\tau}\,dB_t$. We have

$$
\begin{aligned}
\frac{d\mathbf{P}}{d\mathbf{W}^\tau} &= \exp\left(\frac{1}{\sqrt{\tau}}\int_0^t (\Xi + \nabla\Psi)(s, X_s)\,dB_s - \frac{1}{2\tau}\int_0^t |\Xi + \nabla\Psi|^2(s, X_s)\,ds\right) \\
&= \exp\left(\frac{1}{\sqrt{\tau}}\int_0^t \nabla\Psi(s, X_s)\,dB_s - \frac{1}{2\tau}\int_0^t |\nabla\Psi|^2(s, X_s)\,ds\right) \\
&\quad \cdot \exp\left(\frac{1}{\sqrt{\tau}}\int_0^t \Xi(s, X_s)\,dB_s - \frac{1}{2\tau}\int_0^t (|\Xi|^2 + 2\langle\Xi, \nabla\Psi\rangle)(s, X_s)\,ds\right) \\
&= \exp\left(\frac{1}{\tau}\left(\Psi(0, X_0) - \Psi(1, X_1) + \int_0^t \left(\partial_s\Psi - \frac{1}{2}|\nabla\Psi|^2 + \frac{\tau}{2}\Delta\Psi\right)(s, X_s)\,ds\right)\right) \\
&\quad \cdot \exp\left(\frac{1}{\sqrt{\tau}}\int_0^t \Xi(s, X_s)\,dB_s - \frac{1}{2\tau}\int_0^t (|\Xi|^2 + 2\langle\Xi, \nabla\Psi\rangle)(s, X_s)\,ds\right) \\
&= \exp\left(\frac{1}{\tau}\left(\Psi(0, X_0) - \Psi(1, X_1) + \sqrt{\tau}\int_0^t \Xi(s, X_s)\,dB_s\right)\right), \\
&\quad \cdot \exp\left(\frac{1}{\tau}\int_0^t \left(\partial_s\Psi - \frac{1}{2}|\Xi + \nabla\Psi|^2 + \frac{\tau}{2}\Delta\Psi\right)(s, X_s)\,ds\right)
\end{aligned}
\tag{16}
$$

where the first line follows from the Girsanov theorem, (Øksendal, 2010, Thm. 8.6.6) and the third line follows from Itô's formula, (Øksendal, 2010, Thm. 4.2.1). Letting $\mathbf{W}^{\Xi,\tau}$ be the measure induced by the process $dZ_t = -\Xi(t, Z_t)\,dt + \sqrt{\tau}\,dB_t$, we have

$$\frac{d\mathbf{W}^\tau}{d\mathbf{W}^{\Xi,\tau}} = \exp\left(\frac{1}{2\tau}\int_0^t |\Xi|^2\,ds - \frac{1}{\sqrt{\tau}}\int_0^t \Xi\,dB_s\right) \tag{17}$$

by Girsanov. Combining (16) and (17) yields (15). The claim follows. $\qquad\square$

The next result is the variational characteristic of the SDE.

**Theorem C.4** (analogous to (Lavenant et al., 2024, Thm. 2.1))**.** *Suppose* $(g, \Xi, \mathcal{C}_\Psi)$ *is* $\mathcal{C}_\Psi$*-ensemble observable. Let* $\Xi : [0,1] \times \mathcal{X} \to \mathcal{X}$ *be a smooth function and* $\Psi : [0,1] \times \mathcal{X} \to \mathbb{R}$ *be a smooth potential. Let* $\mathbf{P}$ *be the law of the SDE*

$$dX_t = -\Xi(t, X_t)\,dt - \nabla\Psi(t, X_t)\,dt + \sqrt{\tau}\,dB_t$$

*with initial condition* $\mathbf{P}_0 \in \mathcal{P}(\mathcal{X})$ *such that* $H(\mathbf{P}_0|\text{vol}) < +\infty$*. If* $\mathbf{R} \in \mathcal{P}(\Omega)$ *is such that* $g_\sharp \mathbf{R}_t = g_\sharp \mathbf{P}_t$ *for all* $t \in [0,1]$*, we have*

$$H(\mathbf{P}|\mathbf{W}^{\Xi,\tau}) \leq H(\mathbf{R}|\mathbf{W}^{\Xi,\tau})$$

*with equality if and only if* $\mathbf{P} = \mathbf{R}$*.*

The argument follows that of (Lavenant et al., 2024) with our ensemble observable assumption and reference measure. Here, the proof is the same, but now we use the fact that our reference measure "cancels out" the stochastic integral, e.g. see Proposition C.3.

*Proof.* Let $\mathbf{P}$ be the law of the solution of (1) and suppose $\mathbf{R} \in \mathcal{P}(\Omega)$ is another path measure such that $H(\mathbf{R}|\mathbf{W}^{\Xi,\tau}) < +\infty$. Let $p, r \in L^1(\Omega, \mathbf{W}^{\Xi,\tau})$ denote the Radon-Nikodym derivative of $\mathbf{P}, \mathbf{R}$ with respect to $\mathbf{W}^{\Xi,\tau}$, respectively. By strict convexity of $x \mapsto x\log x$, we have

$$r\log r - p\log p \geq (1 + \log p)(r - p),$$

$\mathbf{W}^{\Xi,\tau}$-almost everywhere, with equality if and only if $r = p$. Integrating with respect to $\mathbf{W}^{\Xi,\tau}$, we see

$$H(\mathbf{R}|\mathbf{W}^{\Xi,\tau}) - H(\mathbf{P}|\mathbf{W}^{\Xi,\tau}) \geq \mathbb{E}_{\mathbf{R}}[1 + \log p] - \mathbb{E}_{\mathbf{P}}[1 + \log p]. \tag{18}$$

Using Proposition C.3, we have

$$\mathbb{E}_{\mathbf{R}}[1 + \log p] = \mathbb{E}_{\mathbf{R}}\left[ 1 + \log\left( \frac{d\mathbf{P}_0}{d\mathrm{vol}} \right)(X_0) + \frac{\Psi(0, X_0) - \Psi(1, X_1)}{\tau} \right.$$
$$\left. + \frac{1}{\tau}\int_0^1 \left( \partial_s \Psi - \frac{1}{2}|\nabla\Psi|^2 - \langle \Xi, \nabla\Psi \rangle + \frac{\tau}{2}\Delta\Psi \right)(s, X_s)\, ds \right].$$

By definition of ensemble observability, if $g_\sharp \mathbf{R}_t = g_\sharp \mathbf{P}_t$ for all $t \in [0, 1]$, this implies $\mathbf{R}_t = \mathbf{P}_t$ for all $t \in [0, 1]$. Because this expression only depends on the temporal marginals of $\mathbf{R}$ as the Radon-Nikodym derivative of $\mathbf{P}$ with respect to $\mathbf{W}^{\Xi,\tau}$ does not contain a stochastic integral, the right-hand side of (18) vanishes if $g_\sharp \mathbf{R}_t = g_\sharp \mathbf{P}_t$ for all $t \in [0, 1]$. This concludes the proof. $\qquad \square$

## C.2 THE MAIN TECHNICAL RESULT: THEOREM C.5

**Theorem C.5** (analogous to (Lavenant et al., 2024, Thm. 2.7)). *Fix $\lambda > 0$ and assume we have the following:*

*(i) For every $T \in \mathbb{N}$, we have a sequence of ordered observation times $\{t_i^T\}_{i=1}^T$; a sequence of data $\hat{\rho}_i^T$ (a collection of $T$ probability measures on $\mathcal{Y}$); and a sequence of non-negative weights $\{\omega_i^T\}_{i=1}^T$.*

*(ii) There exists a $\mathcal{P}(\mathcal{Y})$-valued continuous curve $\overline{\rho} \in C([0,1] : \mathcal{P}(\mathcal{Y}))$ such that the following weak convergence holds: for all continuous functions $a : [0,1] \times \mathcal{Y} \to \mathbb{R}$,*

$$\lim_{T \to +\infty} \sum_{i=1}^T \omega_i^T \int_{\mathcal{X}} a(t_i^T, x)\hat{\rho}_i^T(dx) = \int_0^1 \int_{\mathcal{X}} a(t, x)\overline{\rho}(dx)\, dt.$$

*For each $T$, let $\mathbf{R}^T \in \mathcal{P}(\Omega)$ be the unique minimizer of*

$$\mathbf{R} \mapsto F_T(\mathbf{R}) := \tau H(\mathbf{R}|\mathbf{W}^{\Xi,\tau}) + \frac{1}{\lambda}\sum_{i=1}^T \omega_i^T \mathrm{DF}(g_\sharp \mathbf{R}_{t_i^T}, \hat{\rho}_i^T). \tag{19}$$

*Then as $T \to +\infty$, the sequence $\{\mathbf{R}^T\}$ converges weakly on $\mathcal{P}(\Omega)$ to the unique minimizer of*

$$\mathbf{R} \mapsto F(\mathbf{R}) := \tau H(\mathbf{R}|\mathbf{W}^{\Xi,\tau}) + \frac{1}{\lambda}\int_0^1 \mathrm{DF}(g_\sharp \mathbf{R}_t, \overline{\rho}_t).$$

Before proving this theorem, we state the following result that is immediate from the non-negativity of our data-fitting term.

**Fact C.6** (Non-negativity). *With the assumptions of Theorem C.5, the functionals $F_T$ and $F$ are bounded from below by $0$.*

*Proof of Theorem C.5.* The argument follows that of (Lavenant et al., 2024). Let $\mathbf{R}$ be a minimizer of $F$ and $\mathbf{R}^T$ be the minimizer of $F_T$. By optimality of the minimizers, we have $\mathcal{G}_0\mathbf{R} = \mathbf{R}$ and $\mathcal{G}_0\mathbf{R}^T = \mathbf{R}^T$. Using Proposition C.15, we can find a sequence $\tilde{\mathbf{R}}^T$ that converges weakly to $\mathbf{R}$ as $T \to +\infty$ such that

$$F(\mathbf{R}) \geq \limsup_{T \to +\infty} F_T(\tilde{\mathbf{R}}^T) \geq \limsup_{T \to +\infty} \min_{\mathcal{P}(\Omega)} F_T = \limsup_{T \to +\infty} F_T(\mathbf{R}^T).$$

In particular, the sequence is bounded, which by Fact C.6, implies the sequence $H(\mathbf{R}^T|\mathbf{W}^{\Xi,\tau})$ is bounded. Then from the compactness of the sublevel sets of the entropy, we have a limit point $\widehat{\mathbf{R}}$ of the sequence $\{\mathbf{R}^T\}$. Using the optimality of $\mathbf{R}$ and Proposition C.16, we see

$$F(\mathbf{R}) \leq F(\mathcal{G}_0\widehat{\mathbf{R}}) \leq \liminf_{T \to +\infty} F_T(\mathbf{R}^T).$$

Thus, we have equalities everywhere, so

$$F(\mathbf{R}) = \limsup_{T \to +\infty} F_T(\tilde{\mathbf{R}}^T) = \lim_{T \to +\infty} F_T(\mathbf{R}^T).$$

Then we see

$$F_T(\tilde{\mathbf{R}}) - F_T(\mathbf{R}^T) = F_T(\tilde{\mathbf{R}}^T) - \min_{\mathcal{P}(\Omega)} F_T$$

converges to 0 as $T \to +\infty$. By (Lavenant et al., 2024, Lem. B.3), relative entropy is 1-convex with respect to the total variation, i.e. if $p, q, r$ are three probability measures,

$$H\left(\frac{p+q}{2}\bigg|r\right) \leq \frac{1}{2}H(p|r) + \frac{1}{2}H(q|r) - \frac{1}{2}\|p-q\|_{\mathrm{TV}}^2.$$

Since the data-fitting term is also convex, the full objective $F_T(\cdot)$ is 1-convex with respect to the total variation. By a classic strong convexity argument, since $F_T(\tilde{\mathbf{R}})$ converges to the minimum value $\min_{\mathcal{P}(\Omega)} F_T$ achieved at $\mathbf{R}^T$, $\|\tilde{\mathbf{R}}^T - \mathbf{R}^T\|_{\mathrm{TV}}$ must also converge to 0 as $T \to +\infty$. Recall that TV convergence is stronger than weak convergence. Then, using the weak convergence of $\tilde{\mathbf{R}}^T$ to $\mathbf{R}$, we see that $\mathbf{R}^T$ converges weakly to $\mathbf{R}$ as $T \to +\infty$. This concludes the proof. $\square$

The remainder of Appendix C.2 is dedicated towards proving Theorem C.5.

### C.2.1 HEAT FLOW AND REGULARIZATION OF THE MARGINALS

Recall that we use $\Phi_s$ to denote the heat flow with width $s$. We use the heat flow to regularize the marginals. First, we have the following result showing that the density of $\Phi_s \mathbf{R}_t$ is continuous jointly in $t$ and $x$.

**Proposition C.7** (analogous to (Lavenant et al., 2024, Prop. 2.12)). *Let $s > 0$. There exists a constant $C$ depending only on $\mathcal{X}$ and $\Xi$ for which the following hold:*

(i) *For each $\mathbf{R} \in \mathcal{P}(\Omega)$, its heat flow regularization $\Phi_s \mathbf{R}_t$ has density $\rho^{(s)}(t, \cdot)$ (with respect to the volume measure) that satisfies for all $t \in [0,1], x \in \mathcal{X}$, we have*

$$\rho^{(s)}(t, x) \geq \frac{1}{C_s}.$$

(ii) *For all $t_1, t_2 \in [0, 1]$ and $x_1, x_2 \in \mathcal{X}$, we have*

$$|\rho^{(s)}(t_1, x_1) - \rho^{(s)}(t_2, x_2)| \leq C\left(\sqrt{\tau}\sqrt{H(\mathbf{R}|\mathbf{W}^{\Xi, \tau}) + C} + C\tau\sqrt{|t_1 - t_2|} + d_{\mathcal{X}}(x_1, x_2)\right).$$

*Proof.* The first estimate is directly from (Lavenant et al., 2024). The second estimate follows from (Lavenant et al., 2024) and the following proposition. $\square$

**Proposition C.8** (analogous to (Lavenant et al., 2024, Lem. 2.13)). *There exists a constant $C$ depending only on $\mathcal{X}$ and $\Xi$ such that for each $\mathbf{R} \in \mathcal{P}(\Omega)$,*

$$\mathbb{E}_{\mathbf{R}}[d_{\mathcal{X}}(X_{t_1}, X_{t_2})^2] \leq C\left(H(\mathbf{R}|\mathbf{W}^{\Xi, \tau}) + C + C\sigma^2\right)\sigma^2|t_1 - t_2|.$$

*Proof.* The argument follows that of (Lavenant et al., 2024). For any $\eta > 0$, using the dual representation of entropy with the function $X \mapsto \eta d_{\mathcal{X}}(X_{t_1}, X_{t_2})$, we have

$$\eta \mathbb{E}_{\mathbf{R}}[d_{\mathcal{X}}(X_{t_1}, X_{t_2})^2] \leq H(\mathbf{R}|\mathbf{W}^{\Xi, \tau}) + \log \mathbb{E}_{\mathbf{W}^{\Xi, \tau}}[\exp(\eta d_{\mathcal{X}}(X_{t_1}, X_{t_2})^2].$$

Using an upper bound on the heat kernel from (Li & Yau, 1986, Cor. 3.1) and that $\|\Xi\|_{L^\infty} < +\infty$, we have the following bound on the transition probability for $\mathbf{W}^{\Xi, \tau}$:

$$p_\tau(x, y, t) \leq \frac{Ce^{\|\Xi\|_{L^\infty}^2}}{(\tau t)^{d/2}} \exp\left(C\tau t - \frac{d_{\mathcal{X}}(x, y)^2}{C\tau t}\right).$$

Then the remainder of the argument of the proof of (Lavenant et al., 2024, Prop. 2.13) yields the desired result. $\square$

### C.2.2 Heat flow and entropy on the space of paths

We introduce an auxiliary variational problem in which all the temporal marginals are fixed.

**Definition 3.** *Let $\rho \in C([0,1] : \mathcal{P}(\mathcal{X}))$ be a $\mathcal{P}(\mathcal{X})$-valued continuous curve with respect to the weak topology. Define the problem $\mathcal{A}_\tau(\rho)$ to be*

$$\mathcal{A}_\tau(\rho) := \inf_{\mathbf{R} \in \mathcal{P}(\Omega)} \{\tau H(\mathbf{R}|\mathbf{W}^{\Xi,\tau}) \mid \forall t \in [0,1], g_\sharp \mathbf{R}_t = g_\sharp \rho_t\}.$$

*We use the convention that $\mathcal{A}_\tau(\rho) = +\infty$ if the above problem has no admissible competitor.*

Using a dual representation of $\mathcal{A}$, we can use PDE theory to solve this problem. First, we give a martingale characterization of a class of stochastic processes:

**Proposition C.9.** *Suppose $\tilde{\mathbf{W}}^{\Xi,\tau}$ is the law of the SDE $dX_t = -\Xi \, dt + \sqrt{\tau} \, dB_t$ with arbitrary initial distribution. Let $\varphi : [0,1] \times \mathcal{X} \to \mathbb{R}$ be a smooth function. Then, the process whose value at $t \in [0,1]$ is given by*

$$\exp\left(\frac{1}{\tau}\left(\varphi(t,X_t) - \varphi(0,X_0) - \int_0^t \left[\partial_s \varphi + \frac{1}{2}|\nabla\varphi|^2 - \langle\Xi,\nabla\varphi\rangle + \frac{\tau}{2}\Delta\varphi\right](s,X_s)\,ds\right)\right) \quad (20)$$

*is an $\mathcal{F}_t$-martingale under $\tilde{\mathbf{W}}^{\Xi,\tau}$.*

*Proof.* By Assumption C.2 on $\Xi$, the process

$$M_t^\varphi = \varphi(t,X_t) - \varphi(0,X_0) - \int_0^t \left[\partial_s \varphi - \langle\Xi,\nabla\varphi\rangle + \frac{\tau}{2}\Delta\varphi\right](s,X_s)\,ds \quad (21)$$

is an $\mathcal{F}_t$-martingale under $\tilde{\mathbf{W}}^{\Xi,\tau}$ by (Øksendal, 2010, Thm 8.3.1). We calculate the quadratic variation $\langle M^\varphi\rangle_t$ similar to (Hsu, 2008, Prop. 1.3.1). First, we have

$$\varphi(t,X_t)^2 = \varphi(0,X_0)^2 + M_t^{\varphi^2} + \frac{1}{2}\int_0^t [-\langle\Xi,\nabla\varphi^2\rangle + \Delta\varphi^2](s,X_s)\,ds$$

$$= \varphi(0,X_0)^2 + M_t^{\varphi^2} + \frac{1}{2}\int_0^t [-2\varphi\langle\Xi,\nabla\varphi\rangle + \Delta\varphi^2](s,X_s)\,ds.$$

Using Itô's formula, we have

$$\varphi(t,X_t)^2 = \varphi(0,X_0)^2 + 2\int_0^t \varphi(s,X_s)d\varphi^2(s,X_s) + \langle M^\varphi\rangle_t$$

$$= \varphi(0,X_0)^2 + 2\int_0^t \varphi(s,X_s)dM_s^\varphi + \int_0^t \varphi(s,X_s)[-\langle\Xi,\nabla\varphi\rangle + \Delta\varphi](s,X_s)ds + \langle M^\varphi\rangle_t.$$

Equating the bounded variation parts, we see

$$\langle M^\varphi\rangle_t = \frac{1}{2}\int_0^t \left[-2\varphi\langle\Xi,\nabla\varphi\rangle + \Delta\varphi^2 + 2\varphi\langle\Xi,\nabla\varphi\rangle - \varphi\Delta\varphi\right](s,X_s)\,ds$$

$$= \frac{1}{2}\int_0^t \left[\Delta\varphi^2 - \varphi\Delta\varphi\right](s,X_s)\,ds$$

$$= \int_0^t |\nabla\varphi(s,X_s)|^2\,ds.$$

Then, (20) is the exponential martingale of $M_t^\varphi$. $\qquad\square$

Here, recall that Assumption C.2 ensures that (21) is a martingale. Otherwise, it is only a *local martingale* and we would need to check the $L^1$ convergence of the stopped process with an increasing sequence of stopping times that goes to $+\infty$. This is a standard argument in stochastic calculus, e.g. see (Øksendal, 2010).

Now we give the dual representation mentioned above.

**Proposition C.10** (analogous to (Lavenant et al., 2024, Prop. 2.15)). *Let $\rho \in C([0,1] : \mathcal{P}(\mathcal{X}))$ be a $\mathcal{P}(\mathcal{X})$-valued continuous curve. We have*

$$\mathcal{A}_\tau(\rho) = \tau H(\rho_0 | \text{vol})$$
$$+ \sup_\varphi \left\{ -\int_\mathcal{X} \varphi(0,x)\rho_0(dx) - \int_0^1 \int_\mathcal{X} \left( \partial_t \varphi + \frac{1}{2}|\nabla\varphi|^2 - \langle \Xi, \nabla\varphi \rangle + \frac{\tau}{2}\Delta\varphi \right) \rho_t(dx)\,dt \right\},$$

*where the supremum is taken over all $\varphi \in C^2([0,1] \times \mathcal{X})$ such that $\varphi(1,\cdot) = 0$.*

*Proof.* The argument follows that of (Lavenant et al., 2024). From the duality result of (Arnaudon et al., 2020, Prop. 2.3), we have

$$\mathcal{A}_\tau(\rho) = \tau H(\rho_0|\text{vol}) + \tau \sup_\psi \left\{ \int_0^1 \int_\mathcal{X} \psi \rho_t(dx)\,ds - \int_\mathcal{X} \left[ \log \mathbb{E}_{\mathbf{W}^{\Xi,\tau,x}} \exp\left( \int_0^1 \psi\,dt \right) \rho_0(dx) \right] \right\},$$

where $\mathbf{W}^{\Xi,\tau,x}$ is the measure such that $\mathbf{W}_0^{\Xi,\tau} = \delta_x$ and the supremum is taken over $\psi \in C([0,1] \times \mathcal{X})$. Here, the characterization holds because the argument in (Arnaudon et al., 2020) only requires the reference measure to be uniform at all marginals.

Let $-\psi := \frac{1}{\tau}(\partial_t\varphi + \frac{1}{2}|\nabla\varphi|^2 - \langle \Xi, \nabla\varphi \rangle + \frac{\tau}{2}\Delta\varphi)$ for some smooth $\varphi$ satisfying the terminal condition $\varphi(1,x) = 0$. By Proposition C.9, we see

$$\mathbb{E}_{\mathbf{W}^{\Xi,\tau,x}} \left[ \exp\left( -\frac{\varphi(0,X_0)}{\tau} + \int_0^t \psi(t,X_t)\,dt \right) \right] = 1.$$

As $X_0 = 0$ under $\mathbf{W}^{\Xi,\tau,x}$, we have

$$\int_\mathcal{X} \left[ \log \mathbb{E}_{\mathbf{W}^{\Xi,\tau,x}} \exp\left( \int_0^t \psi(t,X_t)\,dt \right) \right] = \int_\mathcal{X} \left[ \log e^{\tau^{-1}\varphi(0,x)} \right] \rho_0(dx)$$
$$= \frac{1}{\tau} \int_\mathcal{X} \varphi(0,x)\,\rho_0(dx).$$

The remaining argument of (Lavenant et al., 2024) follows through. $\qquad\square$

The main idea is that there is a contraction of $\mathcal{A}_\tau$ under the heat flow, which we can think of as a space-time counterpart of the contraction of entropy under the heat flow.

**Proposition C.11** (analogous to (Lavenant et al., 2024, Prop. 2.16)). *Let $\rho \in C([0,1] : \mathcal{P}(\mathcal{X}))$ be a $\mathcal{P}(\mathcal{X})$-valued continuous curve and for $s \geq 0$, define the new curve $\rho^{(s)} : t \mapsto \Phi_s\rho_t$. Let $K$ be a lower bound on the Ricci curvature of the manifold $\mathcal{X}$. Then, for any $s \geq 0$, we have*

$$\mathcal{A}_\tau(\rho^{(s)}) \leq e^{-2Ks}\mathcal{A}_\tau(\rho).$$

*Proof.* Consider the dual formulation in Proposition C.10. If $\varphi : [0,1] \times \mathcal{X} \to \mathbb{R}$ is a $C^2$ function with boundary condition $\varphi(1,\cdot) = 0$, then by the self-adjointness property of the heat semigroup, we have

$$\int_\mathcal{X} \varphi(0,\cdot)\rho_0^{(s)} + \int_0^1 \int_\mathcal{X} \left( \partial_t\varphi + \frac{1}{2}|\nabla\varphi|^2 - \langle \Xi, \nabla\varphi \rangle + \frac{\tau}{2}\Delta\varphi \right) \rho_t^{(s)}\,dt$$
$$= \int_\mathcal{X} \{\Phi_s\varphi\}(0,\cdot)\rho_0 + \int_0^1 \int_\mathcal{X} \left( \partial_t\Phi_s\varphi + \frac{1}{2}\Phi_s|\nabla\varphi|^2 - \Phi_s\langle \Xi, \nabla\varphi \rangle + \frac{\tau}{2}\Delta\Phi_s\varphi \right) \rho_t\,dt,$$

where $\Phi_s\partial_t\varphi = \partial_t\Phi_s\varphi$ by Schwarz's theorem and $\Phi_s\Delta\varphi = \Delta\Phi_s\varphi$ by simple calculation. By properties of the *carré du champ* operator (Bakry et al., 2013, Cor. 3.3.19) and expanding out the inner product, we see that $\langle \Phi_s\Xi, \nabla\Phi_s\varphi \rangle \leq e^{-2Ks}\Phi_s\langle \Xi, \nabla\varphi \rangle$. Thus, letting $\tilde\varphi = e^{2Ks}\Phi_s\varphi$ and $\tilde\Xi = e^{2Ks}\Phi_s\Xi$, we have

$$-\int_\mathcal{X} \varphi(0,\cdot)\rho_0^{(s)} - \int_0^1 \int_\mathcal{X} \left( \partial_t\varphi + \frac{1}{2}|\nabla\varphi|^2 - \langle \Xi, \nabla\varphi \rangle + \frac{\tau}{2}\Delta\varphi \right) \rho_t^{(s)}\,dt$$
$$\leq -e^{-2Ks} \left[ \int_\mathcal{X} \tilde\varphi(0,\cdot)\rho_0 - \int_0^1 \int_\mathcal{X} \left( \partial_t\tilde\varphi + \frac{1}{2}|\nabla\tilde\varphi|^2 - \langle \tilde\Xi, \nabla\tilde\varphi \rangle + \frac{\tau}{2}\Delta\tilde\varphi \right) \rho_t\,dt \right]$$
$$\leq e^{-2Ks}[\mathcal{A}_\tau(\rho) - \tau H(\rho_0|\text{vol})],$$

where the last inequality is due to Proposition C.10. Taking a supremum over $\varphi$, we see that

$$\mathcal{A}_\tau(\rho^{(s)}) \leq e^{-2Ks}\mathcal{A}_\tau(\rho) + \tau\left[H(\Phi_s\rho_0|\mathrm{vol}) - e^{-2Ks}H(\rho_0|\mathrm{vol})\right].$$

By (Lavenant et al., 2024, Eq. B.3), the second term in the right-hand side is always non-positive, so the claim follows. $\qquad\square$

Next, we define the regularizing operator $\mathcal{G}_s$ that acts at the level of laws on the space of paths.

**Definition 4.** *For each $\mathbf{R} \in \mathcal{P}(\Omega)$ with $H(\mathbf{R}|\mathbf{W}^{\Xi,\tau}) < +\infty$ and for each $s \geq 0$, define*

$$\mathcal{G}_s(\mathbf{R}) := \underset{\tilde{R}\in\mathcal{P}(\Omega)}{\arg\min}\{H(\tilde{\mathbf{R}}|\mathbf{W}^{\Xi,\tau}) \mid \forall t \in [0,1], g_\sharp\tilde{\mathbf{R}}_t = g_\sharp\Phi_s\mathbf{R}_t\}.$$

*That is, among all probability distributions on the space of paths whose marginals in hidden space coincide with $t \mapsto g_\sharp\Phi_s\mathbf{R}_t$, the measure $\mathcal{G}_s(\mathbf{R}) \in \mathcal{P}(\Omega)$ is the one with the smallest entropy.*

Note that $\mathcal{G}_s(\mathbf{R})$ is well-defined because thanks to Proposition C.11, $\mathcal{A}_\tau((\Phi_s\mathbf{R}_t)_t) \leq e^{-2Ks}\mathcal{A}_\tau((\mathbf{R}_t)_t) \leq e^{-2Ks}H(\mathbf{R}|\mathbf{W}^{\Xi,\tau}) < +\infty$, so the minimization problem has admissible solutions. Since sublevel sets of entropy are compact, there exists a minimizer, and from strict convexity of the entropy functional, it is unique. Now note that

$$\mathcal{A}_\tau((\Phi_s(\mathbf{R}_t)_t) = H(\mathcal{G}_s(\mathbf{R}|\mathbf{W}^{\Xi,\tau})).$$

This gives us the following result.

**Proposition C.12** (analogous to (Lavenant et al., 2024, Prop. 2.18)). *For each $\mathbf{R} \in \mathcal{P}(\Omega)$ such that $H(\mathbf{R}|\mathbf{W}^{\Xi,\tau}) < +\infty$, we have the following:*

 *(i) For any $s \geq 0$, $H(\mathcal{G}_s(\mathbf{R})|\mathbf{W}^{\Xi,\tau}) \leq e^{-2Ks}H(\mathcal{G}_0(\mathbf{R})|\mathbf{W}^{\Xi,\tau}) \leq e^{-2Ks}H(\mathbf{R}|\mathbf{W}^{\Xi,\tau})$.*

 *(ii) $\mathcal{G}_s(\mathbf{R})$ converges to $\mathcal{G}_0(\mathbf{R})$ weakly as $s \to 0^+$.*

*Proof.* The argument follows that of (Lavenant et al., 2024). The first property is a rewriting of Proposition C.11 together with the definition of $\mathcal{G}_s$ and $\mathcal{A}_\tau$. The second property follows from our observability assumption and an analogous argument to that of the proof of (Lavenant et al., 2024, Prop. 2.18). Consider the following sequential characterization. Let $\{s_n\}_{n\in\mathbb{N}}$ be a sequence with $s_n \to 0$ as $n \to +\infty$. By the contraction estimate in (i) and that the Ricci curvature is bounded from below, we know that $H(\mathcal{G}_{s_n}\mathbf{R}|\mathbf{W}^{\Xi,\tau})$ is uniformly bounded in $n$. Let $\tilde{\mathbf{R}}$ be any limit point of $\mathcal{G}_{s_n}\mathbf{R}$. Notice that this limit point exists due to the compactness of the sublevel sets of $H(\cdot|\mathbf{W}^{\Xi,\tau})$.

We show that $\tilde{\mathbf{R}} = \mathcal{G}_0\mathbf{R}$ by a standard analytic argument. We consider a subsequence (which we do not relabel) $\mathcal{G}_{s_n}\mathbf{R}$ that converges to $\tilde{\mathbf{R}}$ as $n \to +\infty$. The marginals of $\tilde{\mathbf{R}}$ agree with those of $\mathbf{R}$ as we easily see that the marginals of $\mathcal{G}_{s_n}\mathbf{R}$ are the $\{\Phi_{s_n}\mathbf{R}_t\}_{t\in[0,1]}$, and $\Phi_{s_n}f \to f$ in $L^1(\mathcal{X}, \mathrm{vol})$ as $s_n \to 0$. Then, using the lower semi continuity of entropy, the definition of $\mathcal{G}_{s_n}$, and the contraction estimate for $\mathcal{A}$, we have

$$\begin{aligned}H(\tilde{\mathbf{R}}|\mathbf{W}^{\Xi,\tau}) &\leq \liminf_{n\to+\infty} H(\mathcal{G}_{s_n}\mathbf{R}|\mathbf{W}^{\Xi,\tau})\\ &= \liminf_{n\to+\infty} \mathcal{A}_\tau((\Phi_{s_n}\mathbf{R}_t)_t)|\mathbf{W}^{\Xi,\tau})\\ &\leq \liminf_{n\to+\infty} e^{-2Ks_n}\mathcal{A}_\tau((\mathbf{R}_t)_t) = \mathcal{A}_\tau((\mathbf{R}_t)_t).\end{aligned}$$

This shows that $\tilde{\mathbf{R}} = \mathcal{G}_0\mathbf{R}$, which concludes the proof. $\qquad\square$

### C.2.3 THE DATA-FITTING TERM

We recall the definition of the data-fitting term here:

$$\begin{aligned}\mathrm{DF}^\sigma(g_\sharp\mathbf{R}_{t_i^T}, \hat{\rho}_i^T) &:= \int_\mathcal{Y} -\log\left[\int_\mathcal{X} \exp\left(-\frac{\|g(x)-y\|^2}{2\sigma^2}\right) d\mathbf{R}_{t_i^T}(x)\right] d\hat{\rho}_i^T(y)\\ &:= H(\hat{\rho}_i^T|g_\sharp\mathbf{R}_{t_i^T} * \mathcal{N}_\sigma) + H(\hat{\rho}_{t_i^T}) + C,\end{aligned}$$

where $\mathcal{N}_\sigma$ is the Gaussian kernel. First, we have the following result, which is immediate from properties of entropy.

**Proposition C.13.** *The function $r \mapsto \mathrm{DF}(r, p)$ is convex and lower semi continuous on $\mathcal{P}(\mathcal{X})$.*

We will require a quantitative control on the effect of the heat flow on the data-fitting term.

**Proposition C.14** (analogous to (Lavenant et al., 2024, Prop. 2.22)). *Assume $g : \mathcal{X} \to \mathcal{Y}$ is measure preserving.[9] Let $p, r \in \mathcal{P}(\mathcal{X})$. There exists a constant $C > 0$ depending only on $\mathcal{X}$, $g$, and $\sigma$ such that for every $s > 0$,*

$$\mathrm{DF}(g_\sharp \Phi_s r, g_\sharp p) \leq \mathrm{DF}(g_\sharp r, g_\sharp p) + s \cdot C.$$

Above, for simplification of the notation, we pushforward both parameters of the data-fitting term by $g$. This makes the argument below much cleaner.

*Proof.* The argument follows that of (Lavenant et al., 2024), but it is much simpler due to our different data-fitting term. In particular, we do not need a bound on the Fisher information. By an abuse of notation, denote $r \in L^1(\mathcal{X}, \mathrm{vol})$ the density of $r$ with respect to vol. Denote $r(s, \cdot)$ to be the density of $\Phi_s r$ with respect to $r$. It satisfies the heat equation

$$\frac{\partial r}{\partial s} = \Delta r.$$

Then, we have

$$
\begin{aligned}
\frac{d}{ds} \mathrm{DF}(g_\sharp \Phi_s r, g_\sharp p) &= \frac{d}{ds} \int_{\mathcal{Y}} -\log \left[ \int_{\mathcal{X}} \exp\left( -\frac{\|g(x) - g(y)\|^2}{2\sigma^2} \right) r(s, x) \mathrm{vol}(dx) \right] p(y) \mathrm{vol}(dy) \\
&= -\int_{\mathcal{Y}} \log \left[ \int \exp\left( -\frac{\|g(x) - g(y)\|^2}{2\sigma^2} \right) \frac{\partial}{\partial s} r(s, x) \mathrm{vol}(dx) \right] p(y) \mathrm{vol}(dy) \\
&= -\int_{\mathcal{Y}} \log \left[ \int_{\mathcal{X}} \exp\left( -\frac{\|g(x) - g(y)\|^2}{2\sigma^2} \right) \Delta r(s, x) \mathrm{vol}(dx) \right] p(y) \mathrm{vol}(dy) \\
&\leq C \int_{\mathcal{Y}} p(y) \mathrm{vol}(dy) \leq C,
\end{aligned}
$$

where the inequality follows from properties of the Gaussian integral and the fact that $\int \Delta r(s, x) \mathrm{vol}(dx) = 1$. Integrating yields the desired result. $\qquad \square$

### C.2.4 TWO RESULTS ON LIMITS OF FUNCTIONALS

We require two results of the functional $F_T$ defined in (19). We use these for the $\Gamma$-convergence theory required in the proof of Theorem C.5.

**Proposition C.15** (analogous to (Lavenant et al., 2024, Prop. 2.24)). *Use the notation and assumptions of Theorem C.5. Suppose $\mathbf{R} \in \mathcal{P}(\Omega)$ with $F(\mathbf{R}) < +\infty$ and $\mathcal{G}_0 \mathbf{R} = \mathbf{R}$. Then there exists a sequence $\tilde{\mathbf{R}}^T$ which converges weakly to $\mathbf{R}$ as $T \to +\infty$ and*

$$\limsup_{T \to +\infty} F_T(\tilde{\mathbf{R}}^T) \leq F(\mathbf{R}).$$

*Proof.* The argument follows that of (Lavenant et al., 2024). Let $s > 0$. Combining Proposition C.14 for the data-fitting term and Proposition C.12 for the relative entropy on the space of paths, we see that

$$
\begin{aligned}
F(\mathcal{G}_s(\mathbf{R})) &= \tau H(\mathcal{G}_s \mathbf{R} | \mathbf{W}^{\Xi, \tau}) + \frac{1}{\lambda} \int_0^1 \mathrm{DF}(g_\sharp \Phi_s \mathbf{R}_t, \overline{\rho_t}) \, dt \\
&\leq \tau e^{-2Ks} H(\mathbf{R} | \mathbf{W}^{\Xi, \tau}) + \frac{1}{\lambda} \int_0^1 \mathrm{DF}(g_\sharp \mathbf{R}_t, \overline{\rho_t}) \, dt + s \cdot C,
\end{aligned}
$$

so we have

$$\limsup_{s \to 0} F(\mathcal{G}_s \mathbf{R}) \leq F(\mathbf{R}).$$

---

[9]Suppose that $(\mathcal{X}, \lambda_\mathcal{X}), (\mathcal{Y}, \lambda_\mathcal{Y})$ are measure spaces with Lebesgue measure. $g$ is measure preserving if for every Borel set $B \in \mathcal{X}$, $\lambda_\mathcal{X}(A) = \lambda_\mathcal{Y}(g_\sharp A)$.

Now as $-\log[\mathcal{G}_s\mathbf{R}]_t$ is a continuous function of $t$ and $x$ by Proposition C.7, we can use the weak convergence of $\hat{\rho}^T$ to $\overline{\rho}$ to write, for $s > 0$,

$$\lim_{T\to+\infty} \sum_{i=1}^{T} \omega_i^T \mathrm{DF}\left(g_\sharp[\mathcal{G}_s\mathbf{R}]_{t_i^T}, \hat{\rho}_i^T\right) = \int_0^1 \mathrm{DF}(g_\sharp[\mathcal{G}_s\mathbf{R}]_t, \overline{\rho}_t)\, dt.$$

This implies for all $s > 0$, we have $\lim_{T\to+\infty} F_T(\mathcal{G}_s\mathbf{R}) = F(\mathcal{G}_s\mathbf{R})$, so it is sufficient to let $\tilde{\mathbf{R}} := \mathcal{G}_{s_T}\mathbf{R}$ for a sequence $\{s_T\}_{T\geq 1}$ that decays to $0$ sufficiently slowly as $T \to +\infty$. This concludes the proof. $\qquad\square$

**Proposition C.16** (analogous to (Lavenant et al., 2024, Prop. 2.25)). *Use the notation and assumptions of Theorem C.5. For each $T \geq 1$, let $\tilde{\mathbf{R}}^T \in \mathcal{P}(\Omega)$ and assume that it converges weakly to some $\mathbf{R} \in \mathcal{P}(\Omega)$ as $T \to \infty$. Then*

$$F(\mathcal{G}_0\mathbf{R}) \leq \liminf_{T\to+\infty} F_T(\tilde{\mathbf{R}}^T).$$

*Proof.* The argument follows that of (Lavenant et al., 2024). Assume that $\liminf_{T\to+\infty} F_T(\tilde{\mathbf{R}}^T) < +\infty$ otherwise we are done. Then, up to a subsequence (that we do not relabel), we have $\sup_T H(\tilde{\mathbf{R}}^T|\mathbf{W}^{\Xi,\tau}) < +\infty$. Combining Proposition C.14 for the data-fitting term and Proposition C.12 for the relative entropy on the space of paths, we have

$$F_T(\mathcal{G}_s\tilde{\mathbf{R}}^T) = \tau H(\mathcal{G}_s\tilde{\mathbf{R}}^T|\mathbf{W}^{\Xi,\tau}) + \frac{1}{\lambda}\sum_{i=1}^{T} \omega_i^T \mathrm{DF}\left(g_\sharp\Phi_s\tilde{\mathbf{R}}_{t_i^T}^T, \hat{\rho}_i^T\right)$$

$$\leq \tau e^{-2Ks} H(\tilde{\mathbf{R}}^T|\mathbf{W}^{\Xi,\tau}) + \frac{1}{\lambda}\mathrm{DF}\left(g_\sharp\tilde{\mathbf{R}}_{t_i^T}^T, \hat{\rho}_i^T\right) + \frac{sc}{\lambda}.$$

Now we rewrite the above as

$$F_T(\tilde{\mathbf{R}}^T) \geq F_T(\mathcal{G}_s\tilde{\mathbf{R}}^T) - C(s),$$

where

$$C(s) = \tau|e^{-2Ks} - 1|\sup_T H(\tilde{\mathbf{R}}^T|\mathbf{W}^{\Xi,\tau}) + \frac{sc}{\lambda}$$

is upper bounded by a quantity independent of $T$ and $\lim_{s\to 0^+} C(s) = 0$. For the data-fitting term, define the sequence of functions $a_s^T(t,x)$ to be

$$a_s^T(t,x) := -\log\left[\int \exp\left(-\frac{\|g(z) - x\|^2}{2\sigma^2}\right) d\Phi_s\tilde{\mathbf{R}}_t^T(z)\right],$$

which is parametrized by $T$. Notice that from the definition of the data-fitting term, we have

$$\sum_{i=1}^{T} \omega_i^T \mathrm{DF}(g_\sharp\Phi_s\tilde{\mathbf{R}}_{t_i^T}^T, \hat{\rho}_i^T) = \sum_{i=1}^{T} \omega_i^T \int_{\mathcal{X}} a_s^T(t_i^T, x)\hat{\rho}_i^T(dx).$$

For a fixed $s > 0$, the family of functions $a_s^T(t,x)$ indexed by $T$ is uniformly equicontinuous due to $g$ being continuous and Proposition C.7. Then there exists a subsequence (that we do not relabel) that converges uniformly on $[0,1] \times \mathcal{Y}$ as $T \to \infty$ to the function

$$a_s^T(t,x) = -\log\left[\int \exp\left(-\frac{\|g(z) - x\|^2}{2\sigma^2}\right) d\Phi_s\mathbf{R}_t(z)\right]$$

$$= -\log\left[\int \exp\left(-\frac{\|g(z) - x\|^2}{2\sigma^2}\right) d[\mathcal{G}_s\mathbf{R}]_t(z)\right].$$

Using this uniform convergence with the weak convergence of $\hat{\rho}_i^T$ to $\overline{\rho_t}$, we see

$$\lim_{T\to+\infty} \sum_{i=1}^{T} \omega_i^T \mathrm{DF}\left(g_\sharp\Phi_s\mathbf{R}_{t_i^T}^T, \hat{\rho}_i^T\right) = \lim_{T\to+\infty} \sum_{i=1}^{T} \omega_i^T \int_{\mathcal{X}} a_s^T(t_i^T, x)\hat{\rho}_i^T(dx)$$

$$= \int_0^1 \int_{\mathcal{X}} a_s(t,x)\overline{\rho}_t(dx)\, dt$$

$$= \int_0^1 \mathrm{DF}(g_\sharp\mathcal{G}_s\mathbf{R}_t, \overline{\rho}_t)\, dt.$$

Using lower semi continuity of entropy, we have $F(\mathcal{G}_s \mathbf{R}) \leq \liminf_{T \to \infty} F_T(\mathcal{G}_s \tilde{\mathbf{R}}^T)$. Thus, for each $s > 0$, we have

$$\liminf_{T \to +\infty} F_T(\tilde{\mathbf{R}}^T) \geq F(\mathcal{G}_s \mathbf{R}) - C(s).$$

Finally, we use Proposition C.12 to take $s \to 0^+$ using the lower semi continuity of $F$ and the convergence of $\mathcal{G}_s \mathbf{R}$ to $\mathcal{G}_0 \mathbf{R}$ when $s \to 0^+$. □

### C.3 Γ-CONVERGENCE: TAKING $\sigma \to 0, \lambda \to 0$

**Theorem C.17** (analogous to (Lavenant et al., 2024, Thm. 2.9)). *Let* $\mathbf{P} \in \mathcal{P}(\Omega)$ *with* $H(\mathbf{P}|\mathbf{W}^{\Xi,\tau}) < +\infty$. *For each* $\lambda > 0$ *and* $\sigma > 0$, *let* $\mathbf{R}^{\lambda,\sigma}$ *be the minimizer of the functional*

$$\mathbf{R} \mapsto G_{\lambda,\sigma}(\mathbf{R}) := \tau H(\mathbf{R}|\mathbf{W}^{\Xi,\tau}) + \frac{1}{\lambda} \int_0^1 H(\mathbf{P}_t|\mathbf{R}_t * \mathcal{N}_\sigma) \, dt.$$

*Then, as* $h \to 0, \lambda \to 0$, *the measure* $\mathbf{R}^{\lambda,\sigma}$ *converges to the minimizer of* $\mathbf{R} \mapsto H(\mathbf{R}|\mathbf{W}^{\Xi,\tau})$ *among all measures such that* $g_\sharp \mathbf{R}_t = g_\sharp \mathbf{P}_t$ *for all* $t \in [0, 1]$. *Furthermore, if* $\mathbf{P}$ *is the law of the SDE in* (1), *then* $\mathbf{R}^{\lambda,\sigma}$ *converges to* $\mathbf{P}$.

*Proof.* The argument follows that of (Lavenant et al., 2024). First, consider $\mathbf{R} := \mathcal{G}_\sigma \mathbf{P} \in \mathcal{P}(\Omega)$ as a competitor in $G_{\lambda,\sigma}$. Using the contraction estimate given by Proposition C.12, we have

$$\min_{\mathcal{P}(\Omega)} G_{\lambda,\sigma} = G_{\lambda,\sigma}(\mathbf{R}^{\lambda,\sigma}) \leq \tau H(\mathcal{G}_\sigma \mathbf{P}|\mathbf{W}^{\Xi,\tau}) \leq \tau e^{-K\sigma} H(\mathcal{G}_0 \mathbf{P}|\mathbf{W}^{\Xi,\tau}).$$

As $K > -\infty$ and $H(\mathbf{P}|\mathbf{W}^{\Xi,\tau})$ by assumption, we see that $G_{\lambda,\sigma}(\mathbf{R}^{\lambda,\sigma})$ is uniformly bounded in $\lambda$ and $\sigma$. Thus, $H(\mathbf{R}^{\lambda,\sigma}|\mathbf{W}^{\Xi,\tau})$ is uniformly bounded as well. Due to (Lavenant et al., 2024, Prop. B.2), this implies that the family $\mathbf{R}^{\lambda,\sigma}$ belongs to a compact set in the weak topology. Let $\tilde{\mathbf{R}}$ be any limit point in the limit as $\lambda \to 0, \sigma \to 0$. We only need to show that $\tilde{\mathbf{R}} = \mathcal{G}_0 \mathbf{P}$. Note that

$$\tau H(\mathbf{R}^{\lambda,\sigma}|\mathbf{W}^{\Xi,\tau}) \leq G_{\lambda,\sigma}(\mathbf{R}^{\lambda,\sigma}) \leq \tau e^{2K\sigma} H(\mathcal{G}_0 \mathbf{P}|\mathbf{W}^{\Xi,\tau}).$$

By taking $\sigma \to 0$ and using the lower semi continuity of entropy, we see

$$H(\tilde{\mathbf{R}}|\mathbf{W}^{\Xi,\tau}) \leq H(\mathcal{G}_0 \mathbf{P}|\mathbf{W}^{\Xi,\tau}).$$

Now using Fatou's lemma and joint lower semi continuity of the entropy, we have

$$\int_0^1 H(\mathbf{P}_t|\tilde{\mathbf{R}}_t * \mathcal{N}_\sigma) \, dt \leq \liminf_{\lambda \to 0, \sigma \to 0} \int_0^1 H(\Phi_\sigma \mathbf{P}_t|\mathbf{R}_t^{\lambda,\sigma}) \, dt$$

$$\leq \liminf_{\lambda \to 0, \sigma \to 0} \left( \lambda \sup_{\lambda,\sigma} G_{\lambda,\sigma}(\mathbf{R}^{\lambda,\sigma}) \right) = 0.$$

Thus, it follows that $g_\sharp \tilde{\mathbf{R}}_t = g_\sharp \mathbf{P}_t$ for almost every $t$. Therefore, by definition of $\mathcal{G}_0$, we have $\tilde{\mathbf{R}} = \mathcal{G}_0 \mathbf{P}$. This concludes the proof. □

## D REDUCED FORMULATION

### D.1 PROOF OF THEOREM 3.2

We use the following result to prove Theorem 3.2. Here, the statement is identical to that of (Chizat et al., 2022), but we consider a different reference measure.

**Lemma D.1** (analogous to (Chizat et al., 2022, Prop. B.2)). *There exists a constant* $C > 0$ *such that, for any* $\mathbf{R} \in \mathcal{P}(\Omega)$ *and* $t_1^T, \ldots, t_T^T$ *a collection of time instants, it holds*

$$H(\mathbf{R}|\mathbf{W}^{\Xi,\tau}) \overset{(\dagger)}{\geq} H(\mathbf{R}_{t_1^T,\ldots,t_T^T}|\mathbf{W}^{\Xi,\tau}_{t_1^T,\ldots,t_T^T})$$

$$\overset{(*)}{\geq} \sum_{i=1}^{T-1} H(\mathbf{R}_{t_i^T,t_{i+1}^T}|p_{\tau_i}^\Xi(\mathbf{R}_{t_i^T} \otimes \mathbf{R}_{t_{i+1}^T})) - \sum_{i=1}^{T-1} H(\mathbf{R}_{t_i^T}|\mathbf{W}^{\Xi,\tau}_{t_i^T}) + C.$$

*The first inequality* (†) *becomes an equality if and only if*

$$\mathbf{R}(\cdot) = \int_{\mathcal{X}^T} \mathbf{W}^{\Xi,\tau}(\cdot|x_1,\ldots,x_T)\, d\mathbf{R}_{t_1^T,\ldots,t_T^T}(x_1,\ldots,x_T),$$

*where* $\mathbf{W}^{\Xi,\tau}(\cdot|x_1,\ldots,x_T)$ *is the law of* $\mathbf{W}^{\Xi,\tau}$ *conditioned on passing through* $x_1,\ldots,x_T$ *at times* $t_1^T,\ldots,t_T^T$, *respectively. In addition, the second inequality* (∗) *becomes an equality if and only if* $\mathbf{R}$ *is Markovian.*

*Proof.* Using the fact that $\Xi$ is divergence-free and that $\mathbf{W}^{\Xi,\tau}$ has the Markov property, the proof from (Chizat et al., 2022) holds. We provide the full proof for completeness. The first inequality (†) and the equality case follows from the behavior of entropy with respect to a Markov measure under conditioning, e.g. (Léonard, 2012, Eq. 11). In particular, we have

$$H(\mathbf{R}|\mathbf{W}^{\Xi,\tau}) = H(\mathbf{R}_{t_1^T,\ldots,t_T^T}|\mathbf{W}^{\Xi,\tau}_{t_1^T,\ldots,t_T^T})$$
$$+ \int H\left(\mathbf{R}(\cdot|x_1,\ldots,x_T)|\mathbf{W}^{\Xi,\tau}(\cdot|x_1,\ldots,x_T)\right) d\mathbf{R}_{t_1^T,\ldots,t_T^T}(x_1,\ldots,x_T),$$

where the second term vanishes if and only if the conditional distributions $\mathbf{R}(\cdot|x_1,\ldots,x_T)$ follow the law of $\mathbf{W}^{\Xi}$, for $\mathbf{R}_{t_1^T,\ldots,t_T^T}$ almost every $(x_1,\ldots,x_T)$. The second inequality (∗) follows from (Benamou et al., 2019, Lem. 3.4), which states

$$H(\mathbf{R}_{t_1^T,\ldots,t_T^T}|\mathbf{W}^{\Xi,\tau}_{t_1^T,\ldots,t_T^T}) \geq \sum_{i=1}^{T-1} H(\mathbf{R}_{t_i^T,t_{i+1}^T}|\mathbf{W}^{\Xi,\tau}_{t_i^T,t_{i+1}^T}) - \sum_{i=2}^{T-1} H(\mathbf{R}_{t_i^T}|\mathbf{W}^{\Xi,\tau}_{t_i^T}) =: E,$$

with equality if and only if $\mathbf{R}_{t_1^T,\ldots,t_T^T}$ is Markovian. As in (Chizat et al., 2022), we reorganize the terms in $E$.

Without loss of generality, assume that $\mathbf{R}_{t_i^T}$ are absolutely continuous with density $d\mathbf{R}_{t_i^T}(x)/dx :=$ $r_i(x)$ and let $V_{\mathcal{X}}$ be the Lebesgue volume of $\mathcal{X}$. Since $\mathbf{W}^{\Xi,\tau}_{t_i^T}$ is the uniform measure on $\mathcal{X}$ for every $t_i^T$, we have

$$H(\mathbf{R}_{t_i^T}|\mathbf{W}^{\Xi,\tau}_{t_i^T}) = H(\mathbf{R}_{t_i^T}) + \log V_{\mathcal{X}}.$$

Letting $\tau_i := \tau(t_{i+1}^T - t_i^T)$, we also have

$$\mathbf{W}^{\Xi,\tau}_{t_i^T,t_{i+1}^T}(dx,dy) = \frac{1}{V_{\mathcal{X}}} p^{\Xi}_{\tau_i}(x,y)\, dx\, dy.$$

Thus, we see that for any $\mu,\nu \in \mathcal{P}(\mathcal{X})$ with finite differential entropy and $\gamma \in \Pi(\mu,\nu)$, we have

$$H(\gamma|\mathbf{W}^{\Xi,\tau}_{t_i^T,t_{i+1}^T}) = \int \log\left(\frac{d\gamma}{dx \otimes dy}\frac{V_{\mathcal{X}}}{p^{\Xi}_{\tau_i}}\right) d\gamma(x,y)$$
$$= \log V_{\mathcal{X}} + \int \log\left(\frac{d\gamma}{p^{\Xi}_{\tau_i} d(\mu \otimes \nu)}\frac{d\mu}{dx}\frac{d\nu}{dy}\right) d\gamma$$
$$= \log V_{\mathcal{X}} + H(\gamma|p^{\Xi}_{\tau_i}(\mu \otimes \nu)) + H(\mu) + H(\nu),$$

where the last line follows from (Marino & Gerolin, 2020, Lem. 1.6). Now using the fact that $\mathbf{R}_{t_i^T,t_{i+1}^T} \in \Pi(\mathbf{R}_{t_i^T},\mathbf{R}_{t_{i+1}^T})$, we have

$$E = \log V_{\mathcal{X}} + \sum_{i=1}^{T} H(\mathbf{R}_{t_i^T,t_{i+1}^T}|p^{\Xi}_{\tau_i}(\mathbf{R}_{t_i^T} \otimes \mathbf{R}_{t_{i+1}^T})) + \sum_{i=1}^{T-1} H(\mathbf{R}_{t_i^T}),$$

which proves the formula. $\qquad\square$

**Theorem D.2** (Thm. 3.2, restated)**.** *Let* Fit $: \mathcal{P}(\mathcal{Y})^T \to \mathbb{R}$ *be any function and let* $\Xi$ *be bounded and divergence-free.*

*(i) If* $\mathcal{F}$ *admits a minimizer* $\mathbf{R}^*$ *then* $(\mathbf{R}^*_{t_1^T},\ldots,\mathbf{R}^*_{t_T^T})$ *is a minimizer for* $F$.

*(ii) If $F$ admits a minimizer $\boldsymbol{\mu}^* \in \mathcal{P}(\mathcal{X})^T$, then a minimizer $\mathbf{R}^*$ for $\mathcal{F}$ is built as*

$$\mathbf{R}^*(\cdot) = \int_{\mathcal{X}^T} \mathbf{W}^{\Xi,\tau}(\cdot|x_1,\ldots,x_T) \, d\mathbf{R}_{t_1^T,\ldots,t_T^T}(x_1,\ldots,x_T),$$

*where $\mathbf{W}^{\Xi,\tau}(\cdot|x_1,\ldots,x_T)$ is the law of $\mathbf{W}^{\Xi,\tau}$ conditioned on passing through $x_1,\ldots,x_T$ at times $t_1^T,\ldots,t_T^T$, respectively and $\mathbf{R}_{t_1^T,\ldots,t_T^T}$ is the composition of the optimal transport plans $\gamma_i$ that minimize $T_{\tau_i,\Xi}(\boldsymbol{\mu}^{*(i)},\boldsymbol{\mu}^{*(i+1)})$, for $i \in [T-1]$.*

*Proof.* The proof from (Chizat et al., 2022) holds using our transition probability densities and OT plans. We provide it for completeness. First, note that a minimizer $\mathbf{R}^* \in \mathcal{P}(\Omega)$ of $\mathcal{F}(\mathbf{R}) = \text{Fit}(\mathbf{Q}_{t_1^T},\ldots,\mathbf{Q}_{t_T^T}) + \tau H(\mathbf{R}|\mathbf{W}^{\Xi,\tau})$ is of the form in Lemma D.1. Let $\boldsymbol{\mu}^{(i)} := \mathbf{R}^*_{t_i^T}$ be its marginals and $\gamma^{(i)} := \mathbf{R}^*_{t_i^T,t_{i+1}^T}$, which clearly satisfies $\gamma^{(i)} \in \Pi(\boldsymbol{\mu}^{(i)},\boldsymbol{\mu}^{(i+1)})$. Using $C := \log V_{\mathcal{X}}$, we see that

$$\mathcal{F}(\mathbf{R}^*) = \text{Fit}(g_\sharp\boldsymbol{\mu}^{(1)},\ldots,g_\sharp\boldsymbol{\mu}^{(T)}) + \tau \sum_{i=1}^{T-1} H(\gamma^{(i)}|p_{\tau_i}^\Xi(\boldsymbol{\mu}^{(i)}\otimes\boldsymbol{\mu}^{(i+1)})) + \tau H(\boldsymbol{\mu}) + C$$

$$\geq \text{Fit}(g_\sharp\boldsymbol{\mu}^{(1)},\ldots,g_\sharp\boldsymbol{\mu}^{(T)}) + \frac{\tau}{\tau_i} \sum_{i=1}^{T-1} T_{\tau_i,\Xi}(\boldsymbol{\mu}^{(i)},\boldsymbol{\mu}^{(i+1)}) + \tau H(\boldsymbol{\mu}) + C,$$

where the inequality becomes an equality if and only if $\mathbf{R}^*_{t_i^T,t_{i+1}^T} = \gamma^{(i)}$ is optimal in the definition of $T_{\tau_i,\Xi}(\boldsymbol{\mu}^{(i)},\boldsymbol{\mu}^{(i+1)})$. The claim follows. $\qquad\square$

# E   MEAN-FIELD LANGEVIN DYNAMICS

Recall that the MFL dynamics is defined as the solution of (11), which we restate below:

$$\begin{cases} dX_s^{(i)} = -\nabla V^{(i)}[\boldsymbol{\mu}_s](X_s^{(i)}) \, ds + \sqrt{2\tau} \, dB_s^{(i)} + d\Phi_s^{(i)}, & \text{Law}(X_0^{(i)}) = \boldsymbol{\mu}_0^{(i)} \\ \boldsymbol{\mu}_s^{(i)} = \text{Law}(X_s^{(i)}), & i \in [T], \end{cases}$$

where $d\Phi_s^{(i)}$ is the boundary reflection in the sense of the Skorokhod problem. The family of laws $\{\boldsymbol{\mu}_s\}_{s\geq 0}$ of this stochastic process are characterized by the following system of PDEs:

$$\partial_s \boldsymbol{\mu}_s^{(i)} = \nabla \cdot (\nabla V^{(i)}[\boldsymbol{\mu}_s]\boldsymbol{\mu}_s^{(i)}) + \tau \Delta \boldsymbol{\mu}_s^{(i)}, \tag{22}$$

which are coupled via the quantity $\nabla V^{(i)}[\boldsymbol{\mu}_s]$. The link between (11) and (22) follows from the Itô-Tanaka formula, see e.g. (Javanmard et al., 2020, Lem. C.3). This is a multi-species PDE where each of the species $\boldsymbol{\mu}^{(i)}$ attempts to minimize $\frac{\Delta t_i}{\lambda}\text{Fit}^{\lambda,\sigma}(\cdot,\hat{\rho}_i^T) + \tau H$ via a drift-diffusion dynamics, and it is connected to $\boldsymbol{\mu}^{(i-1)}$ and $\boldsymbol{\mu}^{(i+1)}$ via Schrödinger bridges.

## E.1   PROPERTIES OF $G$ AND $F$

We describe some properties of functions $G$ (7) and $F$ (8).

Recall that the *first-variation* of $G : \mathcal{P}(\mathcal{X})^T \to \mathbb{R}$ at $\boldsymbol{\mu}$ is the unique (up to an additive constant) function $V[\boldsymbol{\mu}] \in C(\mathcal{X})^T$ such that for all $\boldsymbol{\nu} \in \mathcal{P}(\mathcal{X})^T$,

$$\lim_{\epsilon\to 0} \frac{1}{\epsilon}[G(1-\epsilon)\boldsymbol{\mu} + \epsilon\boldsymbol{\nu}) - G(\boldsymbol{\mu})] = \sum_{i=1}^T V^{(i)}[\boldsymbol{\mu}](x) \, d(\boldsymbol{\nu} - \boldsymbol{\mu})^{(i)}(x).$$

**Proposition E.1** (analogous to (Chizat et al., 2022, Prop. 3.2))**.** *The function $G$ is convex separately in each of its inputs (but not jointly), weakly continuous and its first-variation is given for $\boldsymbol{\mu} \in \mathcal{P}(\mathcal{X})^T$ and $i \in [T]$ by*

$$V^{(i)}[\boldsymbol{\mu}] = \frac{\delta\text{Fit}}{\delta\boldsymbol{\mu}^{(i)}}[\boldsymbol{\mu}] + \frac{\varphi_{i,i+1}}{t_{i+1}^T - t_i^T} + \frac{\psi_{i,i-1}}{t_i^T - t_{i-1}^T},$$

*and*

$$\frac{\delta \text{Fit}}{\delta \boldsymbol{\mu}^{(i)}}[\boldsymbol{\mu}] : x \mapsto -\frac{\Delta t_i}{\lambda} \int \frac{\mathcal{N}_\sigma(g(x) - y)}{(\mathcal{N}_\sigma * g_\sharp \boldsymbol{\mu}^{(i)})(y)} \, d\hat{\rho}(y),$$

*where $(\varphi_{i,j}, \psi_{i,j}) \in C^\infty(\mathcal{X})$ are the Schrödinger potentials for $T_{\tau_i, \Xi}(\boldsymbol{\mu}^{(i)}, \boldsymbol{\mu}^{(j)})$, with the convention that the corresponding term vanishes when it involves $\psi_{1,0}$ or $\varphi_{T,T+1}$. The function $F$ is jointly convex and admits a unique minimizer $\boldsymbol{\mu}^*$, which has an absolutely continuous density (again denoted by $\boldsymbol{\mu}^*$) characterized by*

$$(\boldsymbol{\mu}^*)^{(i)} \propto e^{-V^{(i)}[\boldsymbol{\mu}^*]/\tau}, \quad \text{for } i \in [T].$$

Here, the Schrödinger potentials are classically $L^1$ by standard (entropic) OT theory, but we can extend them to $C^\infty$ functions, as discussed in Chizat et al. (2022).

*Proof.* The argument is similar to that of (Chizat et al., 2022). The properties of $G$ and its first-variation are clear. In particular, the first-variation of $T_{\tau_i, \Xi}$ follows from the fact that $g$ is smooth and (Santambrogio, 2015, Prop. 7.17), and the first-variation of Fit follows by direct calculation. The convexity of $G$ follows from the convexity of $T_{\tau_i, \Xi}$ and the fact that the pushforward of $g$ is linear. The joint convexity of $F$, its unique minimizer, and the characterization of the minimizer follow directly from the argument in the proof of (Chizat et al., 2022, Prop. 3.2). □

## E.2 NOISY PARTICLE GRADIENT DESCENT

Let $m \in \mathbb{N}$ be the number of particles used in the discretization for each of the time marginals $\boldsymbol{\mu}^{(i)}$. For computation, we approximate the MFL dynamics by running noisy gradient descent on the function $G_m : (\mathcal{X}^m)^T \to \mathbb{R}$ defined as $G_m(\hat{X}) := G(\hat{\boldsymbol{\mu}}_{\hat{X}})$, where

$$\hat{\boldsymbol{\mu}}_{\hat{X}}^{(i)} := \frac{1}{m} \sum_{j=1}^m \delta_{\hat{X}_j^{(i)}}.$$

From (Chizat, 2022, Prop. 2.4), we see that $m\nabla_{X_j^{(i)}} G_m(\hat{X}) = \nabla V^{(i)}[\hat{\boldsymbol{\mu}}_{\hat{X}}](\hat{X}_j^{(i)})$. Thus, this yields the discretization of (11):

$$\begin{cases} \hat{X}_j^{(i)}[k+1] = \hat{X}_j^{(i)}[k] - \eta \nabla V^{(i)}[\hat{\boldsymbol{\mu}}[k]](\hat{X}_j^{(i)}[k]) + \sqrt{2\eta\tau} Z_{j,k}^{(i)}, & \hat{X}_j^{(i)}[0] \overset{i.i.d.}{\sim} \boldsymbol{\mu}_0^{(i)} \\ \hat{\boldsymbol{\mu}}^{(i)}[k] = \frac{1}{m} \sum_{j=1}^m \delta_{\hat{X}_j^{(i)}[k]}, & i \in [T], \end{cases} \quad (23)$$

where $\eta > 0$ is a step-size, the $Z_{j,k}^{(i)}$ are i.i.d. standard Gaussian variables, and all the particles should be projected onto $\mathcal{X}$ at each step if $\mathcal{X}$ has boundaries. The MFL dynamics are recovered in the limit as $m \to \infty$ and $\eta \to 0$ (Suzuki et al., 2023; Nitanda et al., 2022; Chizat, 2022).

Recently, (Suzuki et al., 2023; Chen et al., 2023) have shown a uniform-in-time propagation of chaos for the MFL dynamics: the "distance" between the $m$-particle distribution and the infinite-particle limit is order $O\left(\frac{1}{m}\right)$ for all $t > 0$.

## E.3 EXPONENTIAL CONVERGENCE

**Theorem E.2** (Thm. 3.4, restated). *Assume $\mathcal{X}$ is the $d$-torus. Let $\boldsymbol{\mu}_0 \in \mathcal{P}(\mathcal{X})^T$ be such that $F(\boldsymbol{\mu}_0) < +\infty$. Then for $\tau \geq 0$, there exists a unique solution $(\boldsymbol{\mu}_s)_{s \geq 0}$ to the MFL dynamics (11). Let $\tau > 0$ and assume that $\boldsymbol{\mu}_0$ has a bounded absolute log-density, it holds*

$$F_\tau(\boldsymbol{\mu}_s) - \min F_\tau \leq e^{-Cs}(F_\tau(\boldsymbol{\mu}_0) - \min F_\tau),$$

*where $C = \beta e^{-\alpha/\tau}$ for some $\alpha, \beta > 0$ independently of $\boldsymbol{\mu}$ and $\tau$. Moreover, taking a smooth time-dependent $\tau$ that decays asymptotically as $\tilde{\alpha}/\log s$ for some $\tilde{\alpha} > \alpha$, it holds $F_0(\boldsymbol{\mu}_s) - F_0(\boldsymbol{\mu}^*) \lesssim \log \log s / \log s \to 0$ and $\boldsymbol{\mu}_s$ converges weakly to the min-entropy estimator $\boldsymbol{\mu}^*$.*

*Proof.* As in (Chizat et al., 2022), we simply need to verify the assumptions in (Chizat, 2022, Thm. 3.2). Recall that the objective function is of the form $F = G + \tau H$. The stability and regularity

of the first-variation $V$, (Chizat, 2022, Assumption 1), is immediate from (Chizat et al., 2022, Prop. C.2) and that $g$ is bounded. The convexity of $F_0$ and existence of a minimizer for $F_\tau$, (Chizat, 2022, Assumption 2), follows from Proposition E.1.

For the uniform log-Sobolev inequality (LSI), (Chizat, 2022, Assumption 3), first note that the $i$th component of the first-variation of $F_0$ is given by $V^{(i)}[\boldsymbol{\mu}] + \tau \log \boldsymbol{\mu}^{(i)}$. Define $D := \operatorname{diam} \mathcal{X}$ and $E := \operatorname{diam} g_\sharp \mathcal{X}$, where $D < +\infty$ by assumption and $E < +\infty$ as $g$ is bounded. Note that $\operatorname{osc} V^{(i)}[\boldsymbol{\mu}] < +\infty$ as the gradient formula for $\delta \mathrm{Fit}/\delta \boldsymbol{\mu}^{(i)}$ is non-negative and is bounded by $E e^{E^2/(2\sigma^2)}$ and by (Chizat et al., 2022, App. A, Eq. 17), the Schrödinger potential $\varphi_{i,i+1}$ has an oscillation bounded by

$$\sup_{x,y \in \mathcal{X}} c_{t_i^T, t_{i+1}^T}(x, y) - \inf_{x,y \in \mathcal{X}} c_{t_i^T, t_{i+1}^T}(x, y) \leq D^2/2.$$

Following the argument in the proof of (Chizat, 2022, Thm. 3.3), the probability measure proportional to $e^{-(V^{(i)}[\boldsymbol{\mu}] + \tau \log \boldsymbol{\mu}^{(i)})/\tau}$ satisfies a LSI with constant $\rho \geq \alpha e^{-\beta/\tau}$ for some $\alpha, \beta$ independent of $s, \epsilon, \boldsymbol{\mu}_0$.

Then, (Chizat, 2022, Thm. 3.2) guarantees the exponential convergence with rate $e^{-Cs}$ with $C = 2\tau\rho$. Furthermore, the convergence result with simulated annealing follows from (Chizat, 2022, Thm. 4.1). $\qquad\square$

## F  ADDITIONAL EXPERIMENTS

### F.1  SETTINGS FOR EXPERIMENTS IN MAIN TEXT

All experiments were run on an M1 Macbook Air with 16 GB of RAM. Synthetic experiments take a few minutes to run, and Wikipedia experiments take a few hours to run.

**"Constant velocity" model**  In this experiment, the diffusivity parameter is set at $\tau = 0.05$. Particles are initialized from $X_0 \sim \mathcal{N}(0, 0.1^2 \cdot I)$ and simulated over the time interval $t \in [0, 5]$ with marginals sampled at 5 evenly spaced intervals. Both PO-MFL and FO-MFL are applied using $m = 100$ particles, we observe 32 particles at each time point, and we use a kernel width of $\sigma = 1.0$ for the data-fitting term. The optimization procedure is initialized with $\eta = 0.5$ and continues for 2,000 iterations. The number of Sinkhorn iterations for entropic OT is capped at 500 iterations.

For the crossing paths experiment, the diffusivity parameter is set at $\tau = 0.0005$, and the time interval is $[0, 2.25]$, and marginals are sampled at 10 evenly spaced intervals we use $m = 50$ particles.

**Wikipedia data**  In this experiment, the diffusivity parameter is set at $\tau = 0.001$. Particles are initialized uniformly over the interval $[100, 300]$. We use a kernel width of $\sigma = 1.0$ for the data-fitting term. The optimization is initialized with $\eta = 0.5$ and continues for 2,000 iterations. The number of Sinkhorn iterations for entropic OT is capped at 250 iterations. We scale the data by $1/50$ for stability during optimization.

The three websites we use are:

```
https://ja.wikipedia.org/wiki/%E5%B2%A1%E6%9D%91%E6%98%8E%E7%BE%8E
```

```
https://ja.wikipedia.org/wiki/%E4%B8%89%E5%AE%85%E6%B4%8B%E5%B9%B3
```

```
https://ja.wikipedia.org/wiki/%E5%A5%A5%E5%B1%B1%E4%BD%B3%E6%81%B5
```

### F.2  CIRCULAR MOTION MODEL

In the circular motion experiment, the diffusivity parameter is set at $\tau = 0.0002$. Particles are initialized from $X_0 \sim \mathcal{N}(0, 0.1^2 \cdot I)$ and simulated over the time interval $t \in [0, 3.14]$ with marginals sampled at 15 evenly spaced intervals. Both PO-MFL and FO-MFL are applied using $m = 100$ particles, we observe 32 particles at each time point, and we use a kernel width of $\sigma = 1.0$ for the data-fitting term. The optimization procedure is initialized with $\eta = 0.5$ and continues for 4,000 iterations. The number of Sinkhorn iterations for entropic OT is capped at 500 iterations.

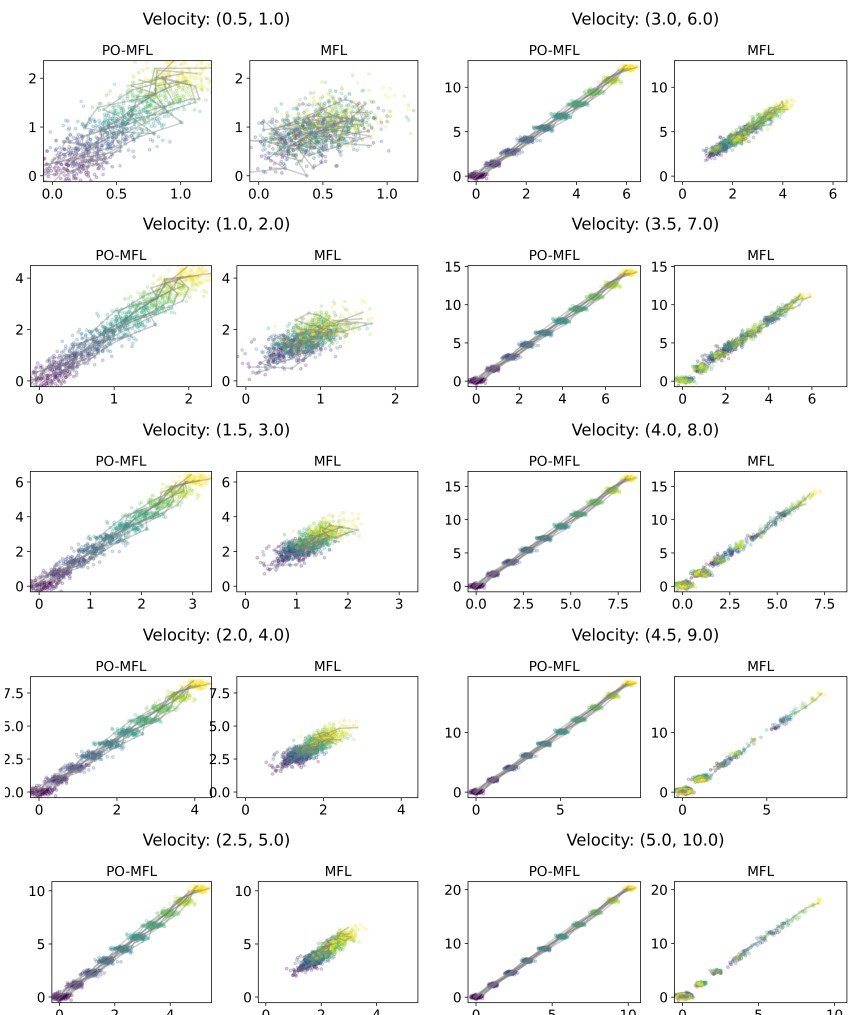

Figure 6: Varying velocity. We see that as the ground truth initial velocity increases, MFL breaks while PO-MFL remains robust.

In our second model, the particles $(\theta, \dot{\theta}, \ddot{\theta}) \in \mathbb{S} \times \mathbb{R}^2$ represent a constant acceleration model on the unit circle, starting from the initial condition $(0, 0.5, 1)$. Here, we use angular velocity and angular acceleration. In this experiment, we only observe the position, e.g. $g(\theta, \cdot, \cdot) = \theta$. In Figure 7a, we show the ground truth with position on the left and angular velocity on the right. In Figure 7b, we show that PO-MFL successfully reconstruct the positions while although FO-MFL converges, it does not recover the ground truth. In Figure 7c, we show that the reconstructed velocity matches that of the ground truth in Figure 7a.

In the following sections, if a parameter is not stated, we assume the same setting of parameters as in the main text.

### F.3 VARYING VELOCITY

Experiments varying the mean of the ground truth initial velocity distribution are shown in Figure 6. At the endpoints, we observe 32 particles, and in the intermediate stages, we observe just 2 particles per time point. Note that in the small velocity regime, although MFL converges, it converges to the wrong distribution.

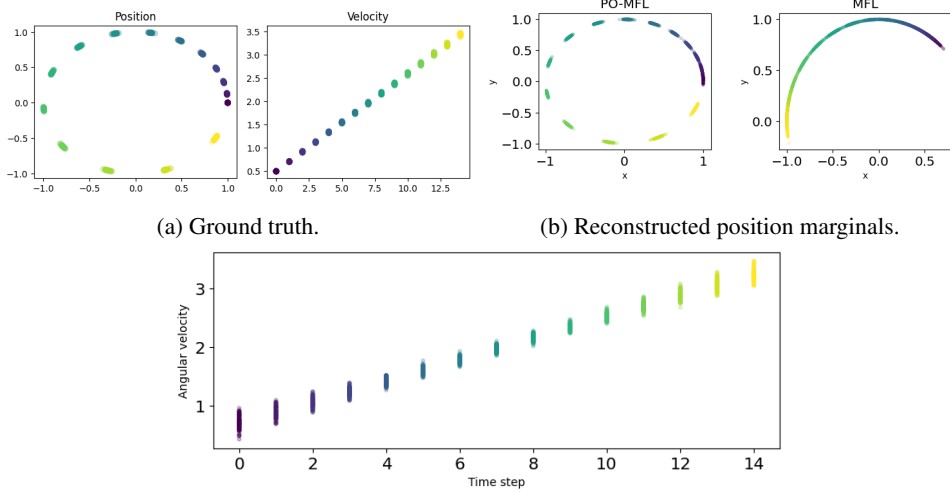

(a) Ground truth.        (b) Reconstructed position marginals.

(c) Reconstructed angular velocity marginals from PO-MFL.

Figure 7: Circular motion model.

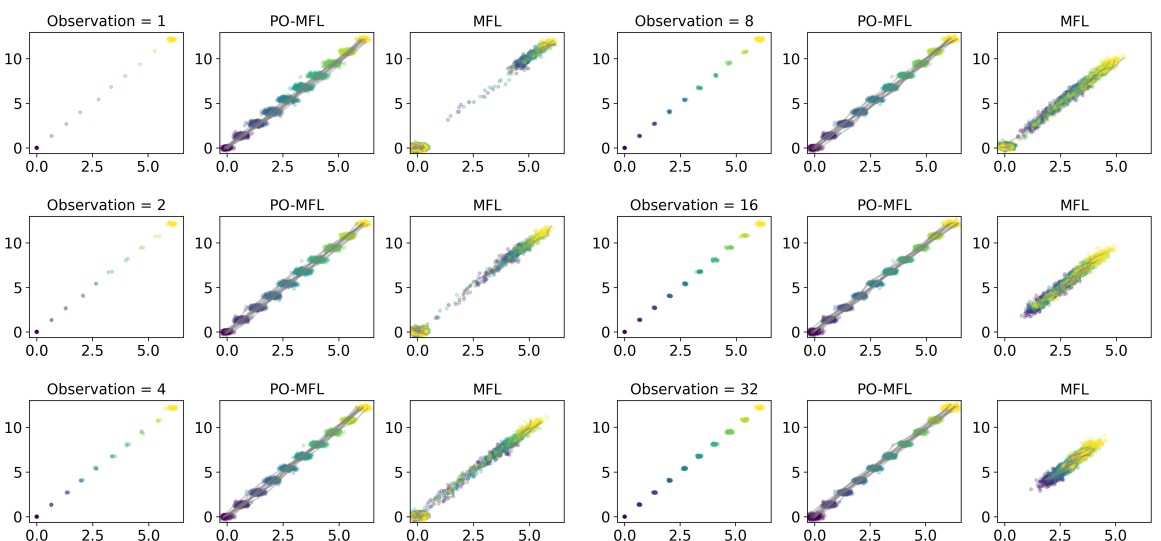

Figure 8: Varying number of observations at the intermediate times. Increasing number of observations improves the optimization.

### F.4 Varying number of observed particles

Figures 8 and 9 show results when the number of observed samples at the intermediate time points are varied (the number of observations at the endpoints is held constant at 32). Here, we try the same settings as above, but now we consider velocity $(\dot{x}, \dot{y}) = (2, 4)$. We try the number of observations $1, 2, 4, 8, 16, 32, 128, 256$. Even in a large number of observation regime, FO-MFL algorithm is not capable of reconstructing the full trajectory, instead clustering around the center of the overall trajectory.

### F.5 Varying temporal sampling density $\Delta t$

In Figure 10, we show results for increasing the density of temporal sampling. At the endpoints, we observe 32 particles, and in the intermediate stages, we observe 2 particles. FO-MFL was sensitive

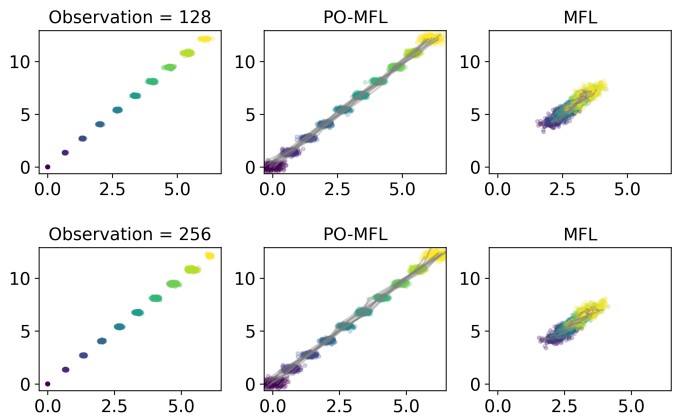

Figure 9: A high observation regime at the intermediate time points.

to hyperparameter values as we needed to try different parameters to get semi-reasonable results for the figure. We used $\sigma = 0.1, m = 25$.

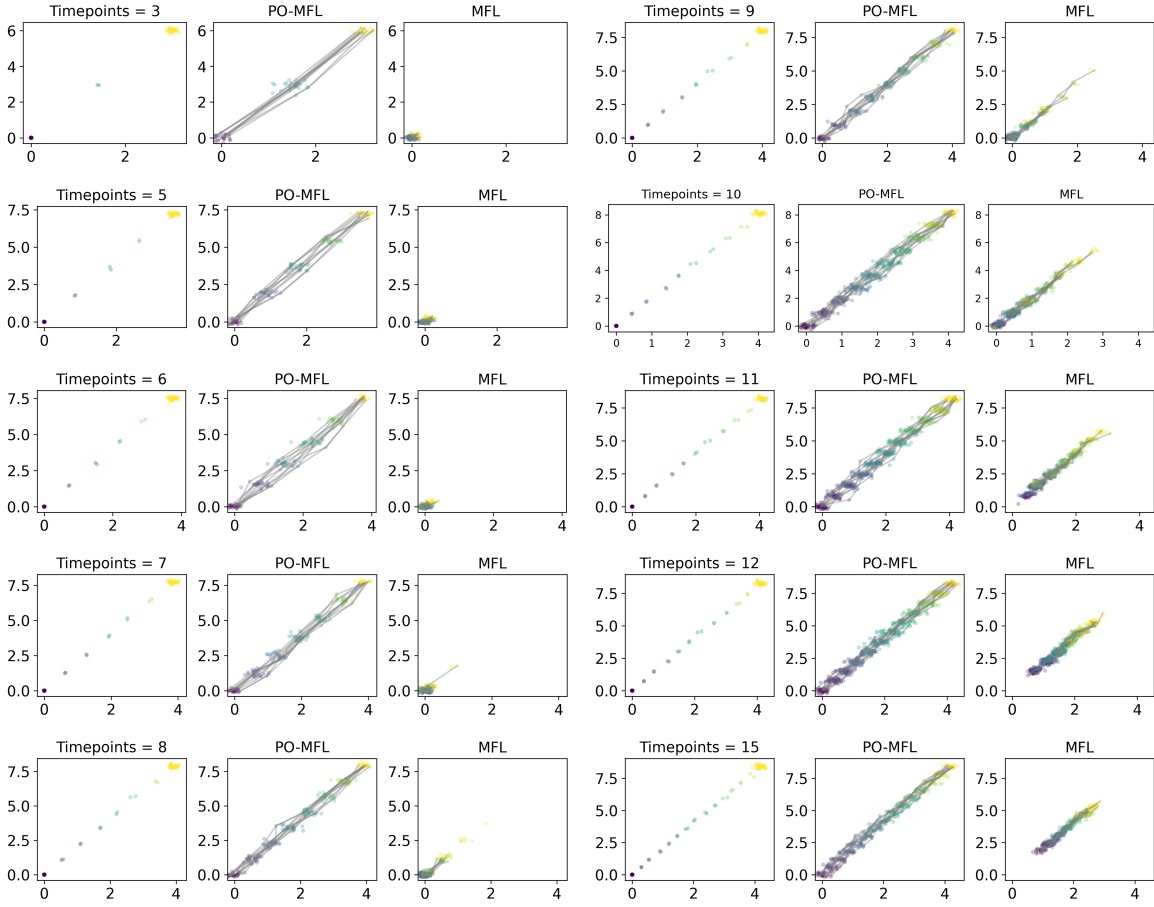

Figure 10: Varying the number of observed time points for fixed time window, i.e. varying $\Delta t$. Note that PO-MFL is always robust. FO-MFL does better with more observations, but the method still tends to collapse inwards because its model suggests that, in expectation, particles should not be moving (as the Brownian motion reference measure has 0 expectation).

