# OpenReview forum: "Partially Observed Trajectory Inference using Optimal Transport and a Dynamics Prior"
_ICLR.cc/2025/Conference — ICLR 2025 Poster_

### Official Review · Reviewer_VsTM · 2024-11-01

**Soundness:** 4
**Presentation:** 3
**Contribution:** 4
**Rating:** 8
**Confidence:** 2

**Summary:**

The paper proposes an extension of existing methods for the trajectory inference problem, focusing on where states are partially observed only through "snapshots" at a given timepoint. These snapshots, for example, capture position but not velocity. They are also unlabelled, for example, you cannot know whether a particle at timepoint t_1 corresponds to a particle at timepoint t_2. The objective is to infer the trajectory distribution of the particles over time.

The authors introduce the PO-MFL (Partially observed mean field Langevin) algorithm to address this challenging problem, developing a deep theoretical contribution that robustly explores their algorithm, including a weak consistency result (section 3.1) showing that their proposed objective function minimises to the true law of the SDE, under some assumptions and in the limit, in ideal conditions. As well as a convergence result (section 3.4), indicating that the PO-MFL algorithm’s approximation of the true trajectory distribution improves over time and eventually stabilises, showing the robustness of PO-MFL.

Finally, the paper grounds their theory in robust experiments - for example, showing empirically that the convergence from Theorem 3.4 holds in Figure 2. Additional experiments show the usefulness of the method, testing robustness across various settings on synthetic and real-world data.

I congratulate the authors on a well constructed paper.

**Strengths:**

1. The paper introduces a well developed framework, extending mean-field Langevin dynamics with latent dynamics modelling for partially observed trajectory inference, making substantial advancements in the field.

2. The paper is theoretically robust - providing a wealth of theory that seeks to capture the applicability and performance of the method, such as the consistency and convergence results.

3. The experiments are varied on synthetic and real-world data, provide insights into the method, and empirically back up theoretical claims.

4. The paper is extremely well written, providing a good background into the subject and gradually introducing concepts to build up to a final goal.

5. The work is framed well within the global context of the problem, highlighting relevant works and providing comparisons and gaps in the literature when needed. Applications of the problem are presented in many cases, showcasing the general applicability of the need for solving the Partially Observed Trajectory Inference problem.

**Weaknesses:**

1. The computational complexity of the algorithm is $O(NTm^2)$, which is not explored further than it is stated, nor is it compared against other methods in the experiments section. Some details on real-world runtime could be useful, due to its (presumably) large scaling cost.

2. Experiments show comparisons to regular MFL, which is not suited for the unobserved case, and therefore there are no comparisons to methods that are suited for this domain. However, I am unfamiliar with the literature, so not sure if there are other methods that can be used in this case.

3. Whilst there is some discussion on the limitation towards model misspecification in the summary section, limitations of the methodology are not discussed in great detail.

**Questions:**

1. How does the computational complexity scale against other methods such as regular MFL?

2. It would improve the strength of the experiments to compare to other methods. Would comparing to something like rejection sampling based methods work? Even with a high computational expense, it could be worth it to compare to as a baseline?

3. Are there any scenarios (within the Partially Observed Trajectory Inference setting) in which PO-MFL will break down? What restrictions does the method have, if you know of any? Are there any limiting factors?

4. Are there any limitations or concerns related to relying on Assumption 2.1? Does the restriction for $\Xi$ to be known, divergence-free and Lipschitz continuous restrict the applicability to some real world systems that have more complex dynamics?

---

> ### Author Response · Authors · 2024-11-19
> **Response**
>
> We thank Reviewer VsTM for their positive review and appreciation of the framework, the theoretical contribution, and writing. In the below, we respond to each question. If anything remains unclear we are happy to discuss in the remaining time!
>
> **The computational complexity of the algorithm is $O(NTm^2)$, which is not explored further than it is stated, nor is it compared against other methods in the experiments section. Some details on real-world runtime could be useful, due to its (presumably) large scaling cost.**
> - The complexity of our algorithm is exactly the same as the baseline. We will note this in the paper.
>
> **Experiments show comparisons to regular MFL, which is not suited for the unobserved case, and therefore there are no comparisons to methods that are suited for this domain. However, I am unfamiliar with the literature, so not sure if there are other methods that can be used in this case.**
> - As far as we are aware, no other work has studied recovering a latent trajectory, so there are not baseline datasets or methods.
> Whilst there is some discussion on the limitation towards model misspecification in the summary section, limitations of the methodology are not discussed in great detail.
> - Some of the model misspecification of $\Xi$ can be swept into $\Psi$, but too much necessarily means we will no longer recover the ground-truth, but we leave this for future work.
>
> **How does the computational complexity scale against other methods such as regular MFL?**
> - See above.
>
> **It would improve the strength of the experiments to compare to other methods. Would comparing to something like rejection sampling based methods work? Even with a high computational expense, it could be worth it to compare to as a baseline?**
> - We are not aware of baselines that compute representative trajectories. In particular, our method’s optimization takes into account the trajectories themselves, e.g. the second term in eq. 7.
>
> **Are there any scenarios (within the Partially Observed Trajectory Inference setting) in which PO-MFL will break down? What restrictions does the method have, if you know of any? Are there any limiting factors?**
> - The key limitation that comes to mind is the need for observability, which is a fundamental constraint of partially observed models, e.g. Kalman filtering.
>
>
> **Are there any limitations or concerns related to relying on Assumption 2.1? Does the restriction for $\Xi$ to be known, divergence-free and Lipschitz continuous restrict the applicability to some real world systems that have more complex dynamics?**
> - In practice, these assumptions do not restrict applicability for real-world systems. $\Xi$ can probably be approximately known, e.g. with some L2 error. Lipschitz continuity is used just for theoretical guarantees, e.g. so that a strong solution of the SDE exists and that the solutions don’t blow up to infinity. Divergence-free is used for simplifying the proofs, and it is likely our method can apply to more general measures, but this would complicate the discussion and analysis.
> - We also note that, via the Helmholtz-Hodge decomposition, any arbitrary differentiable vector field is a sum of an exact component, $\nabla \Psi$, and a divergence-free component, $\Xi$. Thus, our setting encompasses any input differentiable vector field. Knowledge of the divergence-free component is a tool for simplifying the theoretical arguments, as noted above.

---

> > ### Comment · Reviewer_VsTM · 2024-11-21
> > **Thank you for the response**
> >
> > Thank you for your in depth response to my questions.
> >
> > I believe the aspect that your computational complexity is the same as the baseline method is a huge bonus, and should be emphasised in the paper!
> >
> > After reading your response and other reviews here, I will maintain my current positive score, and wish you all the best in the submission.

---

### Official Review · Reviewer_tXu2 · 2024-11-02

**Soundness:** 2
**Presentation:** 3
**Contribution:** 2
**Rating:** 6
**Confidence:** 3

**Summary:**

This work proposes methodology for sampling trajectories of a diffusion which is only partially (but repeatedly) observed at a finite number of time points. At each iteration, the proposed methodology uses optimal-transport ideas to construct a joint distribution over the states of the diffusion at observation times from their marginal distributions (the latter are approximated via samples/"particles"). At the end of each iteration, a mean-field Langevin algorithm is then used to optimise the particle locations.

**Strengths:**

1. **Clarity.** I believe that this work is overall fairly well written (though see some of the comments/questions below, especially with regards to the explanation of some of the experiments).

2. **Novelty/Originality.** The proposed methodology seems new. The main innovation is the extension of a previous approach to allow for a non-zero drift term $\Xi$. This seems fairly moderate, but IMHO sufficient given that it requires non-trivial extensions to the theoretical guarantees which the authors provide.

3. **Quality/Correctness.** This paper comes with a number of rigorously derived theoretical guarantees (though I did not have time to verify all details nor check the derivations in the appendix).

**Weaknesses:**

**Main issues:**


1. **Lack of comparison with simple Monte Carlo alternatives.** I believe the method needs to be compared against a simple Monte Carlo alternative like a particle filter (with backward sampling to obtain trajectories) or a simple MCMC method. The method is currently only compared against an alternative, denoted "MFL", which seems to be so heavily biased that it makes almost anything look good by comparison.

### EDIT: 2024-11-22: As explained in my comment below, Monte Carlo alternatives may likely need to be modified in a not-completely-trivial way to be applicable here. Therefore, I agree that it is fine not to compare against these.


2. **Seemingly high computational cost.** The actual computation time and total number of iterations used in the numerical experiments should be mentioned in the main text. The only mention I found is in Appendix F.1 where it says "[s]ynthetic experiments take a few minutes to run, and Wikipedia experiments take a few hours to run." However, the phrase "a few minutes" is too vague. Further, the synthetic experiments are fairly small (the dimension of $X_t$ seems to be between 2 and 4). I would expect that a simple Monte Carlo alternative (like those mentioned above) would run in mere seconds rather than minutes.


3. **Variance of target distribution seems poorly captured.** As shown in Figure 1c, and commented on in Lines 467--469, the *average* of the sampled velocities seems to match the ground truth. However, the variance seems to be heavily overestimated (or is this just a visual artifact of using scatter plots with different numbers of points?). Additionally, Figure 1a seems to indicate that the variance of the ground truth increases over time but this does not seem to be reflected in the trajectories produced by the proposed method shown in Figure 1b.


4. **Lack of clarity about the real-data experiment.** I find the "Wikipedia traffic data" experiment impossible to understand. In Line 511, the dynamics(?) $x_{t+1} = \theta_1 x_t + \theta_2 x_{t-6}$ are stated. But it is not clear to me how this discrete-time autoregression relates to the SDE fro (1)? That is, what are $\Xi$ and $\tau$ here? Furthermore, are the three websites modelled/estimated separately or somehow jointly (e.g., do the trajectories shown in Figure 5b represent a single website or all three)?


**Minor issues:**

* In Line 32, it should be explained what "unpaired observations" means.

* The term "MFL" appears multiple times throughout the manuscript but sometimes refers to the minimisation algorithm (also used within the proposed methods) and sometimes to the algorithm used as a benchmark in the numerical experiments. I found this confusing, especially because it makes it difficult to find the reference for the benchmark algorithm.

* Figure 2: The tick labels on the 1st axis in the left panel are confusing. Later in the text it is explained that these the final 500 iterations rather than Iterations 1 to 500.

* All axes need to be labelled in all figures. And the 1st-axis labels in Figure 5a are all overlapping and thus unreadable.

* Figures should not appear in the text before they are referenced. This is particularly confusing with Figure 1 which appears on Page 2 but is only referenced/used on Page 9. But there are similar problems with Figures 2 and 3.

* The number of footnotes seems excessive. Many of these, e.g., Footnote 2 an 3, should be included in the text.

* Bibliography: Most entries in the bibliography are missing the journal/conference name. There are also many typos (e.g., missing capitalisation such as "wasserstein", "schroedinger", "brownian", etc).



**Typos:**

* L135: $H(\mu|\nu)$ is defined for probability measures $\mu$ and $\nu$ here but is later used with non-probability measures.

* L144: The last line in the "notation" paragraph seems incomplete since neither $g$ nor $\xi$ have been previously defined and it should probably be stated explicitly that $g_{\sharp\mu}$ denotes the pushforward measure.

* L231: "measurement" -> "measurements"

* Equation 4: Define the "$*$" operator.

* L251: What is $\hat{\mu}_i^{T, h}$?

* L253: "computationally more effective" -> "computationally more efficient"

* L280: Equation 7 is referenced before it appears in the text.

* L289: Shouldn't the first term in the cost function be $-\tau_i$ rather than $-\Delta t_i$?

* L335: Shouldn't "approximate (10)" be replaced by "approximate the first two terms in (10)"?

* L338: There seems to be a space missing at the end of the sentence.

* L362: What is "$\xi$" (i.e., without a superscript) here?

* L367: Use natbib's \citet rather than \citep

* Algorithm 1, Line 8: Is there a "hat" accent missing on $\mathbf{m}$ in the comment?

* Algorithm 1, Lines 8 and 10 (and in the "Require" line): Shouldn't this be "$T$" rather than "$t$"?

**Questions:**

1. What can you say about how this method scales with the dimension of $X_t$?

2. What do you do if $\Xi$ and $\tau$ are unknown?

3. Having a the same diffusion parameter $\tau$ for all components of $X_t$ (e.g., for both the evolution of the positions and of the velocities) seems very limiting. Is there any hope of relaxing this assumption?

4. I didn't understand the "crossing paths" experiment discussed in Lines 477--500 and Figure 4. Can you explain this more clearly?

5. What is the real-world significance of the "Wikipedia traffic data" experiment. In this case, all the traffic data has already been observed. What is the real-world utility of the estimated trajectories?

6. In the caption of Table 1, what does "MCMC" refer to here? I didn't see MCMC methods being mentioned anywhere else.

---

> ### Author Response · Authors · 2024-11-19
> **Response (part 1)**
>
> We thank Reviewer tXu2 for their thorough checking of our paper and detailed response. We address each point one by one.
>
>
> **Lack of comparison with simple Monte Carlo alternatives. I believe the method needs to be compared against a simple Monte Carlo alternative like a particle filter (with backward sampling to obtain trajectories) or a simple MCMC method. The method is currently only compared against an alternative, denoted "MFL", which seems to be so heavily biased that it makes almost anything look good by comparison.**
> - These Particle filter or MCMC methods are, to the best of our knowledge, typically designed to infer the trajectory of a single particle, whereas we are inferring a distribution of trajectories - crucially, without any labels pairing samples from adjacent time points. There is also work (mentioned in the introduction) that infers a trajectory of distributions, but it does not provide an estimate of the coupling between them in order to provide an estimate/samples of the trajectory distribution.
> - The field of trajectory distribution inference is quite new, with a subtle problem statement. As a result, while many existing approaches may seem related, to the best of our knowledge [and implied by Chizat et al.], none of them are applicable to our setting. The closest setting may be multiple target tracking (MTT) using data association methods to match time-adjacent observations (as mentioned in the introduction), but (a) these methods are best suited to disentangling well-separated tracks rather than inferring a crowded trajectory distributions, and (b) to the best of our knowledge do not have limiting consistency theory for inferring the trajectory distribution as we do (particularly in the finite-observations-per-timepoint regime).
> - While indeed MFL severely underperforms in our experiments, this is due to a fundamental failing of the approach when applied to settings with strong drift terms. There are plenty of weakly drifting scenarios without hidden states where MFL performs well, e.g. see Chizat et al. Figure 2.
>
> **Seemingly high computational cost. The actual computation time and total number of iterations used in the numerical experiments should be mentioned in the main text. The only mention I found is in Appendix F.1 where it says "[s]ynthetic experiments take a few minutes to run, and Wikipedia experiments take a few hours to run." However, the phrase "a few minutes" is too vague. Further, the synthetic experiments are fairly small (the dimension of $X_t$ seems to be between 2 and 4). I would expect that a simple Monte Carlo alternative (like those mentioned above) would run in mere seconds rather than minutes.**
> - As our focus was on the extensive theoretical work, we did not make an effort to optimize the code or the implementation. That said, there are many potential directions to streamline the optimal transport portions and make use of GPU architectures and parallelization. For instance, the MFL step can be done in parallel and is a simple operation, and the Sinkhorn step can be accelerated for large numbers of particles using methods such as [SCP21].
> - We also remark that the baseline we compare to has a similar computational cost. We will add this to the discussion and also be more detailed about the computational cost in the revision.
>
> **Variance of target distribution seems poorly captured. As shown in Figure 1c, and commented on in Lines 467--469, the average of the sampled velocities seems to match the ground truth. However, the variance seems to be heavily overestimated (or is this just a visual artifact of using scatter plots with different numbers of points?). Additionally, Figure 1a seems to indicate that the variance of the ground truth increases over time but this does not seem to be reflected in the trajectories produced by the proposed method shown in Figure 1b.**
> - In the output of PO-MFL, similar to the baseline e.g. see Figure 2 of Chizat et al., we do not deconvolve the Gaussian in time induced by the Wiener measure so that Figure 1b is actually tracking the ground truth Figure 1a with constant noise added, making the inferred distributions wider by a constant. We will update Figure 1 in the revision so that (a) and (b) are consistent. In the meantime, note that the variance of the ground truth in Figure 1a at the end time matches that of the variance of our output.

---

> ### Author Response · Authors · 2024-11-19
> **Part 2**
>
> **Lack of clarity about the real-data experiment. I find the "Wikipedia traffic data" experiment impossible to understand. In Line 511, the dynamics(?) $x_{t+1} = \theta_1 x_t + \theta_2 x_{t-6}$ are stated. But it is not clear to me how this discrete-time autoregression relates to the SDE fro (1)? That is, what are $\Xi$ and $\tau$ here? Furthermore, are the three websites modelled/estimated separately or somehow jointly (e.g., do the trajectories shown in Figure 5b represent a single website or all three)?**
> - We will be more clear about the experiment details. We note that $\Xi$ is just a theoretical framework for putting a known prior on the (potential) latent variables. We will add pseudocode on how this can be applied algorithmically in the revision.
> - For real-world applications, $\tau$ is a parameter we can set. Here, it is $0.001$: see line 1668.
> - The three websites are modeled jointly: they form the ground-truth distribution. Then, the sampled trajectories represent all three.
>
> **Minor issues**
> - We will update these in the revision and we briefly remark on the following:
>
> **In Line 32, it should be explained what "unpaired observations" means.**
> - We will add this in the revision. This formally corresponds to assumption (iv) in Theorem C.1 (line 927)
>
> **The term "MFL" appears multiple times throughout the manuscript but sometimes refers to the minimisation algorithm (also used within the proposed methods) and sometimes to the algorithm used as a benchmark in the numerical experiments. I found this confusing, especially because it makes it difficult to find the reference for the benchmark algorithm.**
> - We will fix this in the revision, using MFL for the optimization method and a different term for the benchmark.
>
> **Figures should not appear in the text before they are referenced. This is particularly confusing with Figure 1 which appears on Page 2 but is only referenced/used on Page 9. But there are similar problems with Figures 2 and 3.**
> - We put Figure 1 on page 2 for intuition on our problem setting. We will add discussion referencing this in the introduction.
>
> **Typos**
>
> **L135: $H(\mu|\nu)$ is defined for probability measures $\mu$ and $\nu$ here but is later used with non-probability measures.**
> - We’re not able to find where we use this for non-probability measures - which equation(s) are you referring to?
>
> **L144: The last line in the "notation" paragraph seems incomplete since neither $g$ nor $\xi$ have been previously defined and it should probably be stated explicitly that $g_\sharp \mu$ denotes the pushforward measure.**
> - We will update this in the revision: e.g. for function $g$
>
> **Equation 4: Define the “$*$” operator.**
> - We will mention that this is the convolution operator.
>
> **L251: What is $\Hat{\mu}_i^{T,h}$?**
> - Thanks for catching this. This is the analogous object as the $\Hat{\rho}_i^{T,h}$, except in latent space. We will update this (or remove it if it is confusing).
>
> **L280: Equation 7 is referenced before it appears in the text.**
> - Thank you for catching this: this should be referring to Equation 5. We will fix this.
>
> **L289: Shouldn't the first term in the cost function be $-\tau_i$ rather than $\Delta \tau_i$?**
> - Yes, thanks for catching this.
>
> **L335: Shouldn't "approximate (10)" be replaced by "approximate the first two terms in (10)"?**
> - Yes, we will update this.
>
> L362: What is "$\xi$" (i.e., without a superscript) here?
> - This is a typo, the full expression should be $\log(p_{\tau_i}(\xi^{\Delta t}(x), y)))$
>
> Algorithm 1, Line 8: Is there a "hat" accent missing on m in the comment?
> - Yes, we will fix this.
>
> Algorithm 1, Lines 8 and 10 (and in the "Require" line): Shouldn't this be “T" rather than "t"?
> - Yes, we will fix this.
>
> Questions
>
> What can you say about how this method scales with the dimension of $X_t$
> - The method has linear dimension dependence.
>
> What do you do if $\Xi$ and $\tau$ are unknown.
> - It is known, e.g. from Kalman filtering, that it is in general impossible to infer latent states without a known dynamics model of some kind. Thus, if $\Xi$ is unknown, we cannot expect to recover the latent trajectories. If $\tau$ is unknown, it may still be possible (e.g by inferring a time-invariant \tau in some way).
>
> Having a the same diffusion parameter $\tau$ for all components of $X_t$ (e.g., for both the evolution of the positions and of the velocities) seems very limiting. Is there any hope of relaxing this assumption?
> - We used $\tau$ being the same for all of the components for ease of analysis, and indeed prior works on mean-field Langevin dynamics similarly make this assumption. Also note that the stationary distribution has a nice structure due to this assumption.
> - It should be possible relax this assumption, but the math would become significantly more complex. Additionally, note that whitening the data and/or scaling the hidden space (combined with appropriate adjustments to Xi) can transform a non-uniform \tau into a uniform \tau regime.

---

> ### Author Response · Authors · 2024-11-19
> **Part 3**
>
> **I didn't understand the "crossing paths" experiment discussed in Lines 477--500 and Figure 4. Can you explain this more clearly?**
> - In this experiment, we use the same assumption for other linear dynamics, that is the velocity is undergoing small brownian motion, i.e. it tends to be relatively constant without large jumps. On the other hand, the fully-observed method prefers particles that have velocity v for half the time, then -v for other half, illustrated by a “V” shaped inferred trajectories. Due to the implied regularization of having a velocity hiddens state, our algorithm successfully yields the true ground-truth distribution with crossing paths and more-constant velocity. We will clarify the discussion.
>
> **What is the real-world significance of the "Wikipedia traffic data" experiment. In this case, all the traffic data has already been observed. What is the real-world utility of the estimated trajectories?**
> - While it is true that the Wikipedia data may not have much real-world utility since it is public, it is a real-data standin for envisioned scenarios, described in the introduction, where pairing observations across time may be impossible or undesirable. For instance, if the data domain involves private medical data trajectories (for instance), e.g. members of a study, not using cross-time pairing information and subsampling users at each time could allow for increased privacy to be maintained (a formal privacy analysis is left to future work). Alternately, as mentioned in the introduction, our approach could reduce the need for longitudinal studies, instead allowing for random sampling of sets of individuals at each time (instead of tracking individuals across large amounts of time, a logistical challenge).
>
> In the caption of Table 1, what does "MCMC" refer to here? I didn't see MCMC methods being mentioned anywhere else.
> - This is a typo, this should be Monte Carlo trials. We will fix this in the revision.
>
> **We hope these responses address the reviewer’s concerns and clarify the impact of our paper. If so, any increase in score towards a positive evaluation would be greatly appreciated.**
>
> [SCP21] Scetbon, Cuturi, Peyre. Low-rank Sinkhorn factorization. https://proceedings.mlr.press/v139/scetbon21a/scetbon21a.pdf

---

> ### Comment · Reviewer_tXu2 · 2024-11-22
>
> Thank you for the explanations!
>
> To the first point from Part 1 of the authors' rebuttal, i.e. about the **comparison with particle filters/MCMC**: It's possible that I'm misunderstanding some aspect of the goal/setting. I think one could still apply particle filters here. But, admittedly, these would not be  off-the-shelf, i.e., they would require more research and therefore it it is fine not to compare against such an approach. I will raise my score.
>
> To the first point from Part 2 of the authors' rebuttal, i.e., about the **setup of the Wikipedia experiment**: I was hoping for a bit more details. For instance my question about the "autoregression" has not been addressed.

---

> > ### Author Response · Authors · 2024-11-26
> > **Wikipedia experiments**
> >
> > The experimental setup of the Wikipedia experiments is as follows. We start by obtaining an extra copy of the time series and "lag" the series by 6 days. This serves as the ground truth. From another collection of websites, we repeat this process and calculate the autocorrelation to obtain the parameters in the equation $x_{t+1} = \theta_1 x_t + \theta_2 x_{t-6}$.
> >
> > In the optimization, we optimize for both copies of the time series, using lines 360-361: $(x_{t+1} - (\theta_1 x_t + \theta_2 z_t))^2$ as the squared cost in the first marginal and $(z_{t+1} - z_t)^2$ in the second marginal, where we use $z_t = x_{t-6}$.
> >
> > We remark that a formal description of $\Xi$ is not necessary for applications per se, but it is useful to provide a concise theoretical treatment in Sections 2-3. The Wikipedia experiment is used to demonstrate that our method can straightforwardly implement real-world phenomena (e.g. autocorrelation) while the baseline cannot. We will clarify this in the camera ready. Please let us know if you have any other questions.

---

### Official Review · Reviewer_mqcs · 2024-11-04

**Soundness:** 3
**Presentation:** 1
**Contribution:** 2
**Rating:** 3
**Confidence:** 2

**Summary:**

This paper proposes a latent variable model for trajectory inference, i.e., inferring a time-indexed collection of particles. We assume a latent state vector $(X_t)$ follows some trajectory driven by an SDE, and project the state vector into the observation space via a function $g$, and the objective is to recover the latent trajectories that give rise to the observations.

To infer the latent trajectories, we formulate a minimum entropy functional (an approximation of the KL-divergence with Gaussian kernel smoothing) that can be optimized by an entropy-regularized gradient flow that can be simulated via a McKean-Vlasov process, i.e., the PO-MFL algorithm.

The paper evaluates PO-MFL empirically via toy examples and Wikipedia traffic data, showing improvements compared to the original MFL.

**Strengths:**

While I can't seem to understand part of the paper, I think the overall approach of applying latent variable models to trajectory inference is is a sound and novel approach. I think trajectory inference for latent variables are also a key open problem in this field, and this paper proposes an approach to solving this problem. I also think that the paper's main idea is technically sound for the parts I am able to understand.

The empirical illustration looks convincing and showcases the validity of the algorithm. While I have some reservations about whether or not it is a fair comparison with MFL, I think the paper's approach of inferring latent trajectories + a push-forward function g is overall more preferable than trajectory inference in the observation space.

**Weaknesses:**

While I think the paper presents an interesting approach to trajectory inference, I find the paper's presentation overall unsatisfactory. The explanations are hard to follow, and the notations are perplexing. I think the lack of readability makes it unsuitable to grant an acceptance for the current version, and it makes it difficult to assess the contribution.

- The paper is hard to follow overall. While there is a "notations" section, new notations appear throughout the rest of the paper. Many notation can be inferred from context, but that seems to defeat the purpose of a notation section.
- Some of the confusing notation include 2 different ways to denote convolving variables with a Gaussian density in eq. 4, e.g., both $\mathcal{N}_\sigma$ and $\hat{\rho}_i^{T, h}$. $\gamma$ is both a multiplicative constant in eq. 3 and part of the OT formulation in eq. 6.
- Some of the theoretical explanations in the paper, such as $C_\Psi$-ensemble observability and the general assumptions on $\mathcal{X}$ and $\mathcal{Y}$ are mentioned in the paper, but they are not really included in the discussion.
- There are a number of "filler" type of sentences about "recall that", "note that" which often are re-statements or clarifications. I think it is helpful to reformulate the explanation in a more cogent way. Perhaps it is a good idea to dwell on the most important concepts of the paper, and relegate some less relevant but still beneficial explanation to the appendix.
- Some notations seem unnecessarily complex. For example, I don't understand why the time variables need to be formulated as $t_T^T$.
- The paper makes many references to previous works in the abstract, which makes it look more like a background overview. I think the abstract needs to be restructured in a more self-contained fashion.
- PO-MFL seems to empirically outperform other trajectory inference methods, and latent variable models are ostensibly harder to infer. But this experimental setting (e.g., Figure 4) seems more beneficial, as the latent velocity variables are easier to infer.

**Questions:**

- I don't understand precisely how optimal transport fits into the framework -- eq. 11 is a Wasserstein gradient flow w.r.t. a convexified functional in eq. 8. Is it possible to simplify the framework via Wasserstein gradient flow?
- The paper mentions an unknown potential function $\Psi$ in eq. 1, but does not mention outside the appendix afterwards. Could you explain if inferring $\Psi$ is left for future work, or if it is omitted somewhere in the later notations?
- I don't understand the explanation at the beginning of Section 3.4. Why do we need to increase the entropy by $\epsilon$ when $G + \tau H$ is already convex?

---

> ### Author Response · Authors · 2024-11-19
> **Response (part 1)**
>
> We thank Reviewer mqcs for their review and view of our approach's novelty. Respectfully, however, we are confused by your score of 3 (reject) which seems to be only based on concerns about the clarity of the presentation (which you have independently rated as poor). While presentation can always be improved, we point out that the other reviewers praised the presentation of the work, e.g. Reviewer tXu2 "I believe that this work is overall fairly well written" and Reviewer VsTM "I congratulate the authors on a well constructed paper". The latent trajectory inference problem is subtle and it is inherently difficult to present these advanced concepts within the 10 page limit in a fully self-contained, casually accessible manner, but we do not believe this should by itself lead to rejection as ICLR has a long history of accepting papers with complex theoretical contributions.
>
> That being said, we will absolutely work hard to improve clarity of our presentation in the upcoming revision. We welcome any further concrete presentation recommendations! - and hope that you consider reevaluating your opinion in light of the above.
>
> We address your concerns one by one.
>
> **Weaknesses**
>
> **The paper is hard to follow overall. While there is a "notations" section, new notations appear throughout the rest of the paper. Many notation can be inferred from context, but that seems to defeat the purpose of a notation section.**
> - We will reorganize the notation section, and make sure that every other introduced notation is defined where it is introduced.
>
> **Some of the confusing notation include 2 different ways to denote convolving variables with a Gaussian density in eq. 4, e.g., both $\mathcal{N}_\sigma$ and $\hat{\rho}_i^{T,h}$. $\gamma$ is both a multiplicative constant in eq. 3 and part of the OT formulation in eq. 6.**
> - For the gaussian convolution, thank you for pointing out this diversity, we will choose a unified way of writing this.
> - We do not use $\gamma$ in eq. 3.
>
> **Some of the theoretical explanations in the paper, such as $C_\Psi$-ensemble observability and the general assumptions on $\mathcal{X}$ and $\mathcal{Y}$ are mentioned in the paper, but they are not really included in the discussion.**
> - Due to the limited space, we opted to detail some of these explanations in the appendix, which we will expand. The ensemble observability is a necessary condition to be able to recover the latent trajectories.
> - The assumptions on $\mathcal{X}$ and $\mathcal{Y}$ are (the most general) requirements for the analytical arguments in Appendix C. As they are technical assumptions rather than contributions, they are naturally discussed once without needing too much reiterating in the rest of the paper.
>
> **There are a number of "filler" type of sentences about "recall that", "note that" which often are re-statements or clarifications. I think it is helpful to reformulate the explanation in a more cogent way. Perhaps it is a good idea to dwell on the most important concepts of the paper, and relegate some less relevant but still beneficial explanation to the appendix.**
> - We were hoping that these sentences would aid the reader’s comprehension due to the dense material. If there are specific places where you felt that this repetition was not helpful, let us know.
>
> **Some notations seem unnecessarily complex. For example, I don't understand why the time variables need to be formulated as $t_T^T$.**
> - This notation was borrowed from Lavenant et al., and is necessary due to the fact that the times t_i^T are a set of size T whose locations depend on T. As T will be taken to infinity, having doubly indexed notation is important.
>
> **The paper makes many references to previous works in the abstract, which makes it look more like a background overview. I think the abstract needs to be restructured in a more self-contained fashion.**
> - We will update this in the revision.
>
> **PO-MFL seems to empirically outperform other trajectory inference methods, and latent variable models are ostensibly harder to infer. But this experimental setting (e.g., Figure 4) seems more beneficial, as the latent velocity variables are easier to infer.**
> - In this experiment, the true latent velocities are constant, which makes them easier to infer.

---

> ### Author Response · Authors · 2024-11-19
> **Part 2**
>
> **Questions:**
>
> **I don't understand precisely how optimal transport fits into the framework -- eq. 11 is a Wasserstein gradient flow w.r.t. a convexified functional in eq. 8. Is it possible to simplify the framework via Wasserstein gradient flow?**
> - This is an astute observation, in fact our method provably reduces to the Wasserstein gradient flow in the mean-field limit (as the number of particles goes to infinity). Using mean-field Langevin has several benefits in this context, for instance, the particle updates are easy to compute independently given the Sinkhorn couplings.
> - In terms of other potential algorithms for doing Wasserstein gradient flow, it may be possible to use some kind of JKO scheme, but we envision that this would not necessarily simplify the framework, since it would still involve the computation of Wasserstein distances and there are many Wasserstein terms with both arguments being optimized.
> - Regardless of the precise numerical algorithm, Optimal Transport is the crown jewel of our method, as it provably enables the coupling of information across time points without needing labeled connections.
>
> **The paper mentions an unknown potential function $\Psi$  in eq. 1, but does not mention outside the appendix afterwards. Could you explain if inferring $\Psi$ is left for future work, or if it is omitted somewhere in the later notations?**
> - The prior works also include this unknown potential function, e.g. see footnote 2 (line 212). All of these works (including ours) do not aim to learn $\Psi$. The works [THW+20, HGJ16, BPKC22] do however. Note that the primary purpose of $\Psi$ is to allow for a portion of the true dynamics to be unknown while still retaining consistency of the estimator, giving significant flexibility to practitioners. As a result, being able to handle $\Psi$ is a key advantage of the framework and a significant part of the theoretical analysis. We will add more discussion on this aspect in the main text.
>
> **I don't understand the explanation at the beginning of Section 3.4. Why do we need to increase the entropy by $\epsilon$ when $G+\tau H$ is already convex?**
> - Mean-field Langevin requires optimization of a convex functional $(G + \tau H)$ and entropy. Chizat et al. originally stated that $G$ itself was not convex, but $G+\tau H$ was - hence epsilon was added to provide a guaranteed level of strong convexity. It turns out, however, that $G$ itself is in fact convex, hence this $\epsilon$ is not necessary. We will update this in the revision.
>
> We hope these responses address the reviewer’s concerns and clarify the impact of our paper. If so, any increase in score towards a positive evaluation would be greatly appreciated.
>
> [THW+20] Tong, Huang, Wolf, Van Dijk, Krishnaswamy. TrajectoryNet: a dynamical optimal transport network for modeling cellular dynamics. https://proceedings.mlr.press/v119/tong20a.html
>
> [HGJ16] Hashimoto, Gifford, Jaakkola. Learning population-level diffusions with generative RNNs. https://proceedings.mlr.press/v48/hashimoto16.html
>
> [BPKC22] Bunne, Papaxanthos, Krause, Cuturi. Proximal optimal transport modeling of population dynamics. https://proceedings.mlr.press/v151/bunne22a.html

---

> ### Author Response · Authors · 2024-11-25
> **Discussion period ending soon**
>
> We have just posted a revision (see above) - please let us know if your assessment of the presentation still stands and/or if you have further suggestions we can incorporate.
>
> Thanks!

---

### Author Response · Authors · 2024-11-25
**Thank you for feedback and revision**

We thank all of the reviewers for their feedback. As promised in the below rebuttals, we have just posted a revision where we've updated some of the figures, accounted for the typos, and fixed the discussion by inlining comments and clarifying some points.

---

### Meta-Review · Area_Chair_hHTR · 2024-12-18

**Metareview:**

This work extends trajectory inference to latent state-space models using partially observed data and specified dynamics (e.g., constant velocity/acceleration). It introduces the PO-MFL algorithm with theoretical guarantees, building on prior stochastic differential equation frameworks. Experiments show robust performance, exponential convergence, and significant improvement over latent-free baselines.

Two reviewers recommended acceptance, noting that the paper makes substantial contributions. One reviewer highlighted the soundness and novelty of applying latent variable models to trajectory inference, stating that the empirical illustrations convincingly demonstrate the algorithm's validity. Another reviewer praised the paper for its clarity and originality.

However, the third reviewer recommended rejection, citing a lack of clear presentation in the paper.

The AC sides with the majority vote, acknowledging that the paper's strengths significantly outweigh its weaknesses, such as poor presentation. The AC strongly encourages the authors to improve the paper's presentation, as they committed to doing in their rebuttal.

**Additional Comments On Reviewer Discussion:**

All reviewers raised a few technical questions, which the authors have addressed satisfactorily. I strongly recommend that the authors incorporate these remarks in the final version.

---

### Decision · Program_Chairs · 2025-01-22

Accept (Poster)